# PolyFormer: Scalable Graph Transformer via Polynomial Attention

## Abstract

Graph Transformers have demonstrated superior performance in graph representation learning. However, many current methods focus on attention mechanisms between node pairs, limiting their scalability and expressiveness on node-level tasks. While the recent NAGphormer attempts to address scalability by employing node tokens in conjunction with vanilla multi-head self-attention, these tokens, which are designed in the spatial domain, suffer from restricted expressiveness. On the other front, some approaches have explored encoding eigenvalues or eigenvectors in the spectral domain to boost expressiveness, but these methods incur significant computational overhead due to the requirement for eigendecomposition. To overcome these limitations, we first introduce node tokens using various polynomial bases in the spectral domain. Then, we propose a tailored polynomial attention mechanism, **PolyAttn**, which serves as a node-wise graph filter and offers powerful representation capabilities. Building on PolyAttn, we present **PolyFormer**, a graph Transformer model specifically engineered for node-level tasks, offering a desirable balance between scalability and expressiveness. Extensive experiments demonstrate that our proposed methods excel at learning arbitrary node-wise filters, showing superior performance on both homophilic and heterophilic graphs, and handling graphs containing up to 100 million nodes.

## 1 Introduction

Recently, Graph Neural Networks (GNNs) have been developed to address a range of graph-related problems, such as node classification (Kipf & Welling, 2017; Velickovic et al., 2018; Hamilton et al., 2017), link prediction (Zhang & Chen, 2018), and graph classification (Xu et al., 2019). GNNs can be generally categorized into two types: spatial-based and spectral-based (He et al., 2021). Spatial-based GNNs often rely on a message passing in the spatial domain to aggregate information, whereas spectral-based GNNs perform graph filtering operations in the spectral domain. Concurrently, the Transformer architecture (Vaswani et al., 2017), characterized by its unique attention mechanisms, has been remarkably successful in fields like natural language processing (Devlin et al., 2019; Floridi & Chiriatti, 2020), computer vision (Dosovitskiy et al., 2021; Liu et al., 2021), and audio applications (Dong et al., 2018; Gulati et al., 2020). To leverage the immense power of Transformer models, the Transformer architecture has been recently adapted to graphs, termed graph Transformer, and showcases enhanced performance in graph representation learning (Kreuzer et al., 2021; Ying et al., 2021; Mialon et al., 2021; Rampásek et al., 2022).

Existing graph Transformers generally consider interactions between nodes when calculating attention and then utilize node representations for downstream tasks such as node classification or graph regression (Min et al., 2022; Sun et al., 2023). However, it is worth questioning whether it is necessary to consider attention on node pairs, regardless of node-level or graph-level tasks. In natural language processing and computer vision, Transformer-based models mainly consider interactions among tokens within a sentence or patches within an image, respectively, rather than implementing attention mechanisms between sentences or images (Vaswani et al., 2017; Dosovitskiy et al., 2021). Typically, attention mechanisms are employed on the sub-units constituting a target object. This approach aims to capture the information exchanges among these sub-units, deriving a representation of the target object. Back to graph Transformers, although applying attention to pairs of nodes is justifiable for graph-level tasks, it appears less rational for tasks centered on individual nodes. In fact, two main limitations emerge from focusing on node-pair interactions on node-level tasks:

First, it leads to quadratic computational complexity (Vaswani et al., 2017), hindering scalability on large-scale graphs. Second, it introduces noise and consequently impairs performance (Dwivedi & Bresson, 2020). Therefore, for node-level tasks, it is reasonable to introduce the concept of node tokens and implement the attention mechanism on it.

**Motivation.** Although NAGphormer (Chen et al., 2023) has attempted to use information from various hops as units to represent each node, it designed node tokens on the spatial domain with vanilla multi-head self-attention, neglecting the spectral information, which compromises performance (Kreuzer et al., 2021). This naturally raises the question: *Can we develop node tokens based on the spectral domain with a tailored attention mechanism to capture spectral information and thereby enhance expressive power?*

On the other hand, though NAGphormer and some other existing graph Transformers (Kreuzer et al., 2021; Rampásek et al., 2022; Bo et al., 2023) have attempted to utilize information from the spectral domain, they often rely on eigendecomposition, which is both computationally expensive and memory-intensive. Conversely, polynomial GNNs utilize truncated polynomials to approximate graph filters, avoiding the need for eigendecomposition. However, a significant drawback of polynomial GNNs is that they use shared polynomial coefficients for all nodes, leading to a node-unified filter that inherently limits the model's expressive power (Guo et al., 2023). Consequently, the second question arises: *For node tokens developed in the spectral domain, is it possible to leverage efficient polynomial approximations while overcoming the disadvantages of polynomial GNNs, i.e., the constraint of node-unified filters?*

**Contribution.** In this work, we provide affirmative answers to previously posed questions. First, we introduce a novel node token based on polynomial bases, along with the specifically designed Polynomial Attention (PolyAttn), *to capture information in the spectral domain effectively.* Both theoretical and empirical analyses confirm that the PolyAttn, combined with polynomial node tokens, operates as a node-wise filter. *This offers greater expressiveness compared to node-unified filters while still avoiding burdensome eigendecomposition.* Building on this, we propose a scalable and expressive graph Transformer termed PolyFormer. By leveraging the newly introduced node token, PolyFormer can utilize mini-batch training, thereby significantly enhancing its scalability. Moreover, the expressiveness of PolyAttn allows the model to excel on a variety of graphs, including both homophilic and heterophilic ones. Through extensive experiments, we empirically validate the performance, scalability, and efficiency advantages of PolyFormer on node-level tasks. We summarize the contributions of this paper as follows:

- We introduce polynomial-based node tokens from the spectral domain and propose a tailored attention mechanism, PolyAttn, which is notably expressive. Utilizing the node token and PolyAttn, we propose PolyFormer, a scalable graph Transformer designed for node-level tasks.

- Theoretically, we demonstrate that PolyAttn functions as a node-wise filter with the designed node token. We also illustrate that multi-head PolyAttn serves as a multi-channel filter. Moreover, We explore the computational complexity tied to the proposed node token and PolyAttn.

- Comprehensive experiments validate that PolyAttn possesses greater expressive power than node-unified filters. Building on PolyAttn, PolyFormer achieves a desirable balance between expressive power and computational efficiency. It demonstrates superior performance on both homophilic and heterophilic datasets and is capable of handling graphs with up to 100 million nodes.

## 2 BACKGROUND

**Notations.** Let us consider a graph $G = (V, E)$, where $V$ is the set of nodes and $E$ is the set of edges. The adjacency matrix is denoted as $\mathbf{A} \in \{0, 1\}^{N \times N}$, where $\mathbf{A}_{ij} = 1$ signifies the existence of an edge between nodes $v_i$ and $v_j$, and $N$ is the total number of nodes in the graph. The normalized Laplacian of the graph is defined as $\hat{\mathbf{L}} = \mathbf{I} - \hat{\mathbf{A}} = \mathbf{I} - \mathbf{D}^{-1/2}\mathbf{A}\mathbf{D}^{-1/2}$. In these equations, $\mathbf{I}$ represents the identity matrix, $\hat{\mathbf{A}}$ denotes the normalized adjacency matrix, and $\mathbf{D}$ is a diagonal degree matrix where $\mathbf{D}_{ii} = \sum_j \mathbf{A}_{ij}$. It is well-established that $\hat{\mathbf{L}}$ is a symmetric positive semidefinite matrix, allowing for decomposition as $\hat{\mathbf{L}} = \mathbf{U}\mathbf{\Lambda}\mathbf{U}^\top = \mathbf{U}\text{diag}(\lambda_0, \cdots, \lambda_{N-1})\mathbf{U}^\top$. Here, $\mathbf{\Lambda}$ is a diagonal matrix composed of eigenvalues $\lambda_i, i \in \{0, \cdots, N-1\}$, and $\mathbf{U}$ consists of the corresponding eigenvectors.

**Graph Filter.** Graph filter serves as a crucial concept in the field of graph signal processing (Isufi et al., 2022). Formally, given an original graph signal matrix or, equivalently, node feature matrix $\mathbf{X} \in \mathbb{R}^{N \times d}$, the filtered signal $\mathbf{Z} \in \mathbb{R}^{N \times d}$ is obtained through the graph filtering operation in the spectral domain:

$$\mathbf{Z} = \mathbf{U}h(\mathbf{\Lambda})\mathbf{U}^\top\mathbf{X}, \tag{1}$$

where $h(\mathbf{\Lambda})$ signifies the graph filter. Notably, this filter can become node-wise when $h(\cdot)$ in Equation 1 is tailored for individual nodes, denoted as $h^{(i)}(\cdot)$ for node $v_i$. Conversely, $h(\cdot)$ is considered channel-wise if there exists a corresponding $h_{(j)}(\cdot)$ for each signal channel $\mathbf{X}_{:,j}, j \in \{0, \cdots, d-1\}$. It is worth noting that directly learning $h(\mathbf{\Lambda})$ necessitates eigendecomposition, which has a time complexity of $O(N^3)$.

**Polynomial GNNs.** To alleviate the computational burden of eigendecomposition, recent studies have introduced Polynomial GNNs for approximating $h(\mathbf{\Lambda})$ using polynomials. These Polynomial GNNs can be implemented with various bases, such as Monomial (Chien et al., 2021), Bernstein (He et al., 2021), Chebyshev (Defferrard et al., 2016; He et al., 2022), Jacobi (Wang & Zhang, 2022), and even optimal bases (Guo & Wei, 2023). Using a specific polynomial basis, the approximated filtering operation can be represented as:

$$\mathbf{Z} = \mathbf{U}h(\mathbf{\Lambda})\mathbf{U}^\top\mathbf{X} \approx \sum_{k=0}^{K} \alpha_k g_k(\mathbf{P})\mathbf{X}, \tag{2}$$

where $\alpha_k$ are the polynomial coefficients for all nodes, $g_k(\cdot), k \in \{0, \cdots, K\}$ denotes a series polynomial basis of truncated order $K$, and $\mathbf{P}$ refers to either the normalized adjacency matrix $\hat{\mathbf{A}}$ or the normalized Laplacian matrix $\hat{\mathbf{L}}$. For example, the filtering operation of GPRGNN (Chien et al., 2021) is $\mathbf{Z} = \sum_{k=0}^{K} \alpha_k \hat{\mathbf{A}}^k \mathbf{X}$, which uses the Monomial basis.

**Transformer.** The Transformer architecture (Vaswani et al., 2017) is a powerful deep learning model that has had a significant impact in multiple domains. The critical component of the Transformer is its attention mechanism. For an input matrix $\mathbf{X} = [\boldsymbol{x}_1, \ldots, \boldsymbol{x}_n]^\top \in \mathbb{R}^{n \times d}$, the attention mechanism transforms $\mathbf{X}$ into $\mathbf{Q}$, $\mathbf{K}$, and $\mathbf{V}$ using learnable projection matrices $\mathbf{W}_Q \in \mathbb{R}^{d \times d'}$, $\mathbf{W}_K \in \mathbb{R}^{d \times d'}$, and $\mathbf{W}_V \in \mathbb{R}^{d \times d'}$ as:

$$\mathbf{Q} = \mathbf{X}\mathbf{W}_Q, \quad \mathbf{K} = \mathbf{X}\mathbf{W}_K, \quad \mathbf{V} = \mathbf{X}\mathbf{W}_V. \tag{3}$$

The output of the attention mechanism is computed as:

$$\mathbf{O} = \text{softmax}\left(\frac{\mathbf{Q}\mathbf{K}^\top}{\sqrt{d}}\right)\mathbf{V}. \tag{4}$$

This attention mechanism can be executed multiple times to produce a multi-head attention mechanism.

**Graph Transformer.** The Transformer architecture has been adapted for graph domains, termed graph Transformers, which have gained significant attention. To integrate graph information into Transformer-based models, multiple techniques have been developed, including the use of spectral information (Kreuzer et al., 2021; Rampásek et al., 2022; Bo et al., 2023), GNNs as auxiliary modules (Mialon et al., 2021; Wu et al., 2021), and other encoding strategies (Ying et al., 2021; Zhao et al., 2021). On another front, scalability challenges persist for graph Transformers. Solutions, such as efficient attention mechanisms (Wu et al., 2022), sampling strategies (Zhang et al., 2022), token-based methods (Chen et al., 2023), and approximation with global nodes (Kuang et al., 2021; Kong et al., 2023), have been employed.

## 3 POLYFORMER

In this section, we introduce our proposed PolyFormer, a scalable graph transformer via polynomial attention. First, we define the concept of node tokens based on polynomial bases. Utilizing these node tokens, we describe our attention mechanism and provide an overview of the model architecture. Finally, we analyze the computational complexity of our model and establish its relationship with the graph filters.

Table 1: Recursive Computing Process of Polynomial Tokens for Different Bases.

| Polynomial Type | Initial Value | Recursive Formula |
|---|---|---|
| Monomial Basis | $\mathbf{H}_0 = \mathbf{X}$ | $\mathbf{H}_k = \hat{\mathbf{A}}\mathbf{H}_{k-1}$ |
| Chebyshev Basis | $\mathbf{H}_0 = \mathbf{X}, \mathbf{H}_1 = \hat{\mathbf{L}}\mathbf{X}$ | $\mathbf{H}_k = 2\hat{\mathbf{L}}\mathbf{H}_{k-1} - \mathbf{H}_{k-2}$ |

## 3.1 POLYNOMIAL TOKEN

Analogous to sentence tokenization in natural language processing, we introduce polynomial tokens for nodes to enhance graph Transformer scalability on node-level tasks.

**Definition 3.1. (Polynomial Token)** *For any node $v_i$ in a graph $G = (V, E)$, the polynomial token of the node is defined as $\boldsymbol{h}_k^{(i)} = (g_k(\mathbf{P})\mathbf{X})_{i,:} \in \mathbb{R}^d, k \in \{0, \cdots, K\}$, where $g_k(\cdot)$ represents a polynomial basis of order $k$, $\mathbf{P}$ is either $\hat{\mathbf{A}}$ or $\hat{\mathbf{L}}$, and $\mathbf{X}$ represents the node features.*

In this work, we employ Monomial and Chebyshev bases for polynomial tokens. These choices offer ease of implementation compared to more complex polynomial bases such as Bernstein or Jacobi. Additionally, the Monomial basis provides a clear spatial interpretation, with $\boldsymbol{h}_k^{(i)} = (\hat{\mathbf{A}}^k\mathbf{X})_{i,:}$ representing the information of the $k$-hop neighborhood from node $v_i$. Meanwhile, the Chebyshev basis exhibits excellent fitting capabilities (Geddes, 1978). Both bases can be computed recursively. Table 1 illustrates the recursive computing process for all nodes in the graph, where $\mathbf{H}_k = [\boldsymbol{h}_k^{(0)}, \cdots, \boldsymbol{h}_k^{(N-1)}]^\top \in \mathbb{R}^{N \times d}$ denotes the matrix consisting of polynomial tokens of order $k$ for $k \in \{0, \cdots, K\}$.

The adoption of polynomial tokens offers several distinct advantages. Firstly, these tokens can be computed recursively. Once computed, they can be reused across epochs during both the training and inference phases, leading to substantial reductions in computational time and memory usage. Furthermore, by incorporating the normalized adjacency or Laplacian matrix $\mathbf{P}$ into the computational process, graph topology information is integrated into node tokens. This integration eliminates the necessity for additional position or structure encodings, such as Laplacian eigenvectors, thereby further enhancing the model's efficiency. Finally, the inherent node-wise independence of these polynomial tokens allows for mini-batch training, enabling us to scale the model to graphs with up to 100 million nodes.

## 3.2 POLYATTN AND POLYFORMER

Given the polynomial tokens associated with each node, PolyFormer employs a tailored attention mechanism to generate node representations. Firstly, we introduce the proposed attention mechanism PolyAttn, tailored for polynomial tokens, which acts as a node-wise filter. Subsequently, we detail the comprehensive architecture of PolyFormer.

**PolyAttn.** In this section, we first detail the process of the proposed PolyAttn for a given node $v_i$. Let us define the token matrix $\mathbf{H}^{(i)}$ for node $v_i$ as $\mathbf{H}^{(i)} = [\boldsymbol{h}_0^{(i)}, \cdots, \boldsymbol{h}_K^{(i)}]^\top \in \mathbb{R}^{(K+1) \times d}$. Initially, the value matrix $\mathbf{V}$ is initialized using the token matrix $\mathbf{H}^{(i)}$. Subsequently, we use an order-specific multi-layer perceptron ($\text{MLP}_j$) to map the $j$-th order token $\boldsymbol{h}_j^{(i)} = \mathbf{H}_{j,:}^{(i)}$ into a hidden space. This step allows for the capture of unique contextual information for each order of polynomial tokens, akin to the function of positional embeddings in standard Transformer architecture.

Upon obtaining the query matrix $\mathbf{Q}$ and the key matrix $\mathbf{K}$ by projecting $\mathbf{H}^{(i)}$ through the learnable matrices $\mathbf{W}_Q$ and $\mathbf{W}_K$, respectively, these matrices are used to compute the attention scores. Notably, our attention mechanism employs the hyperbolic tangent function $\tanh(\cdot)$ instead of the softmax function used in the vanilla Transformer (Vaswani et al., 2017) and NAGphormer (Chen et al., 2023). This is because the softmax function limits the expressive capability of PolyAttn when it functions as a node-wise graph filter. Further clarification will be provided in Proposition 3.2.

Additionally, a node-shared attention bias $\boldsymbol{\beta} \in \mathbb{R}^{K+1}$ is introduced to strike a balance between node-specific and global patterns. Finally, the computed attention scores $\mathbf{S}$ are used to multiply with the value matrix $\mathbf{V}$, resulting in the final output representations. The pseudocode of PolyAttn is provided as Algorithm 1. In practice, we employ multi-head PolyAttn to enhance expressive power. More details are provided in Appendix C.

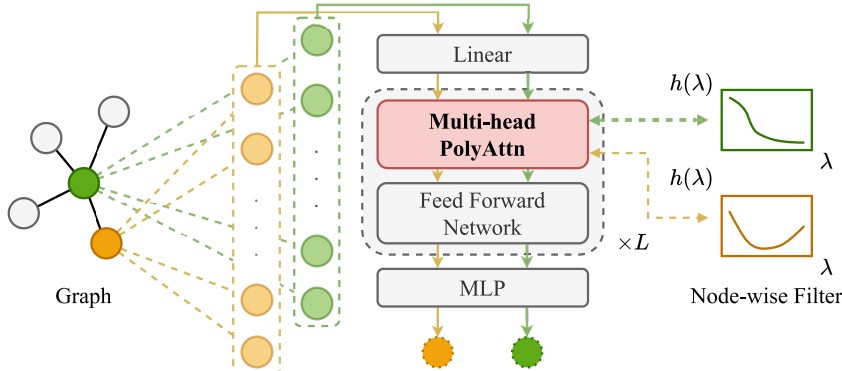

Figure 1: **Illustration of the proposed PolyFormer.** For a given graph, polynomial tokens for each node are computed. These tokens are subsequently processed by PolyFormer, which consists of $L$ blocks. Notably, with the defined polynomial token, PolyAttn within each block functions as a node-wise filter, adaptively learning graph filter specific to each node.

**PolyFormer.** Building upon the attention mechanism designed for polynomial tokens, we introduce the novel graph Transformer model PolyFormer. As illustrated in Figure 1, PolyFormer block is described by the following equations:

$$\mathbf{H}'^{(i)} = \text{PolyAttn}\left(\text{LN}\left(\mathbf{H}^{(i)}\right)\right) + \mathbf{H}^{(i)}, \tag{5}$$

$$\mathbf{H}^{(i)} = \text{FFN}\left(\text{LN}\left(\mathbf{H}'^{(i)}\right)\right) + \mathbf{H}'^{(i)}. \tag{6}$$

Here, LN denotes Layer Normalization, which is implemented before PolyAttn (Xiong et al., 2020). FFN refers to the Feed-Forward Network. Upon obtaining the token matrix $\mathbf{H}^{(i)} \in \mathbb{R}^{(K+1)\times d}$ for node $v_i$ through $L$ PolyFormer blocks, the final representation $\mathbf{Z}_{i,:} \in \mathbb{R}^c$ of node $v_i$ is computed as:

$$\mathbf{Z}_{i,:} = \sigma\left(\left(\sum_{k=0}^{K}\mathbf{H}_{k,:}^{(i)}\right)\mathbf{W}_1\right)\mathbf{W}_2, \tag{7}$$

where $\sigma$ denotes the activation function. The matrices $\mathbf{W}_1 \in \mathbb{R}^{d\times d'}$ and $\mathbf{W}_2 \in \mathbb{R}^{d'\times c}$ are learnable, with $d, d'$ representing the hidden dimensions and $c$ representing the number of node classes.

---

**Algorithm 1:** Pseudocode for PolyAttn

---

**Input:** Token matrix for node $v_i$: $\mathbf{H}^{(i)} = [\boldsymbol{h}_0^{(i)}, \cdots, \boldsymbol{h}_K^{(i)}]^\top \in \mathbb{R}^{(K+1)\times d}$
**Output:** New token matrix for node $v_i$: $\mathbf{H}'^{(i)} \in \mathbb{R}^{(K+1)\times d}$
**Learnable Parameters:** Projection matrix $\mathbf{W}_Q, \mathbf{W}_K \in \mathbb{R}^{d\times d'}$,
               order-wise $\text{MLP}_j (j = 0, \cdots, K)$,
               attention bias $\boldsymbol{\beta} \in \mathbb{R}^{K+1}$

1 Initialize $\mathbf{V}$ with $\mathbf{H}^{(i)}$
2 **for** $j = 0$ to $K$ **do**
3     $\mathbf{H}_{j,:}^{(i)} \leftarrow \text{MLP}_j(\mathbf{H}_{j,:}^{(i)})$
4 $\mathbf{Q} \leftarrow \mathbf{H}^{(i)}\mathbf{W}_Q$ via projection matrix $\mathbf{W}_Q$; $\mathbf{K} \leftarrow \mathbf{H}^{(i)}\mathbf{W}_K$ via projection matrix $\mathbf{W}_K$
5 Compute attention scores $\mathbf{S} \leftarrow \tanh(\mathbf{QK}^\top) \odot \mathbf{B}$, where $\mathbf{B}_{ij} = \beta_j$
6 $\mathbf{H}'^{(i)} \leftarrow \mathbf{SV}$
7 **return** $\mathbf{H}'^{(i)}$     # The representation of node $v_i$ after PolyAttn is $\boldsymbol{Z}_{i,:} = \sum_{k=0}^{K}\mathbf{H}'^{(i)}_{k,:} \in \mathbb{R}^d$.

---

### 3.3 THEORETICAL ANALYSIS

#### 3.3.1 COMPLEXITY

**Computing for Polynomial Tokens.** As previously discussed, the polynomial tokens can be calculated recursively. Each iteration for all nodes involves sparse multiplication with a computational complexity of $O(|E|)$. Thus, the overall complexity is $O(K|E|)$, where $K$ is the truncated order of the polynomial tokens, and $|E|$ is the number of edges in the graph. Importantly, these polynomial tokens can be computed once and reused throughout the training and inference process.

**Complexity of PolyAttn.** Let $d$ denote the hidden dimension of polynomial tokens, and $K$ represent the truncated order. In the context of one layer of PolyAttn, each node involves $(K + 1)$ polynomial tokens in attention computation, resulting in a complexity of $O((K + 1)^2 d)$. With $N$ nodes in the graph and $L$ layers of attention mechanisms, the total time complexity is $O(LN(K + 1)^2 d)$. Notably, in practical situations where $K \ll N$, this signifies a substantial reduction in computational complexity, especially when compared to the $O(LN^2 d)$ complexity of standard Transformer models.

#### 3.3.2 CONNECTION TO SPECTRAL FILTERING

To understand the connection between PolyAttn and graph filters, we give the following theorem and propositions. All proofs are in Appendix B. First, we formally propose that PolyAttn serves as a node-wise filter for polynomial tokens.

**Theorem 3.1.** *With polynomial tokens as input, PolyAttn operates as a node-wise filter. Specifically, for the representation $\mathbf{Z}_{i,:} = \sum_{k=0}^{K} \mathbf{H'}_{k,:}^{(i)}$ of node $v_i$ after applying PolyAttn, we have:*

$$\mathbf{Z}_{i,:} = \sum_{k=0}^{K} \mathbf{H'}_{k,:}^{(i)} = \sum_{k=0}^{K} \alpha_k^{(i)} \left( g_k \left( \mathbf{P} \right) \mathbf{X} \right)_{i,:}. \tag{8}$$

*Here, the coefficients $\alpha_k^{(i)}$ depend not only on the polynomial order $k$ but also on the specific node $v_i$. In other words, PolyAttn performs a node-wise polynomial filter on the graph signals.*

Building on Theorem 3.1 above, we further propose that the multi-head PolyAttn acts as a multi-channel filter.

**Proposition 3.1.** *A multi-head PolyAttn with $h$ heads can be interpreted as partitioning the node representation into $h$ channel groups with dimension $d_h = \frac{d}{h}$ and applying filtering to each group separately. Formally:*

$$\mathbf{Z}_{i,p:q} = \sum_{k=0}^{K} \alpha_{(p,q)k}^{(i)} \left( g_k(\mathbf{P})\mathbf{X} \right)_{i,p:q}. \tag{9}$$

*Here, $\alpha_{(p,q)k}^{(i)}$ denotes the coefficient for order $k$ on channels $p$ to $q$ of node $v_i$'s representation, where $(p, q) = (j \times d_h, (j + 1) \times d_h - 1), j \in \{0, \cdots, h - 1\}$.*

It is worth noting that our chosen activation function, $\tanh(\cdot)$, enables PolyAttn with more powerful expressiveness than the softmax function used in both vanilla Transformer and NAGphormer.

**Proposition 3.2.** *For PolyAttn, which operates as a graph filter, the tanh function endows it with enhanced expressiveness, whereas the softmax function can limit the expressive capability of PolyAttn.*

## 4 EXPERIMENTS

In this section, we conduct comprehensive experiments to evaluate the performance of the proposed PolyAttn and PolyFormer. Specifically, we first evaluate PolyAttn's ability on node-wise filtering using both synthetic and real-world datasets. Then, we execute node classification tasks on both small and large graphs to evaluate the effectiveness and efficiency of PolyFormer.

Table 2: Performance of PolyAttn on Synthetic Datasets ($R^2$ score / the sum of squared error).

| Model (5k para) | Mixed low-pass | Mixed high-pass | Mixed band-pass | Mixed rejection-pass | Low&high-pass | Band&rejection-pass |
|---|---|---|---|---|---|---|
| GCN | 0.9953/2.0766 | 0.0186/39.6157 | 0.1060/14.0738 | 0.9772/10.9007 | 0.6315/86.8209 | 0.8823/128.2312 |
| GAT | 0.9954/2.0451 | 0.0441/38.5851 | 0.0132/14.0375 | 0.9775/10.7512 | 0.7373/61.8909 | 0.9229/83.9671 |
| GPRGNN | 0.9978/0.9784 | 0.9806/0.7846 | 0.9088/1.2977 | 0.9962/1.8374 | 0.8499/35.3719 | 0.9876/13.4890 |
| BernNet | 0.9976/1.0681 | 0.9808/0.7744 | 0.9231/1.0937 | 0.9968/1.5545 | 0.8493/35.5144 | 0.9875/13.6485 |
| ChebNetII | 0.9980/0.8991 | 0.9811/0.7615 | 0.9492/0.7229 | 0.9982/0.8610 | 0.8494/35.4702 | 0.9870/14.1149 |
| PolyAttn (Mono) | 0.9994/0.2550 | 0.9935/0.2631 | 0.9030/1.3798 | 0.9971/1.4025 | 0.9997/0.0696 | 0.9992/0.8763 |
| PolyAttn (Cheb) | **0.9997/0.1467** | **0.9960/0.0148** | **0.9945/0.0782** | **0.9996/0.1949** | **0.9999/0.0118** | **0.9999/0.0416** |

## 4.1 POLYATTN EXPERIMENTS

### 4.1.1 FITTING SIGNALS ON SYNTHETIC DATASETS.

**Synthetic Datasets.** We use images with a resolution of $100 \times 100$ from the Image Processing in Matlab library [1]. Each image can be represented as a 2D regular 4-neighborhood grid graph. The pixel values, ranging from 0 to 1, serve as node signals. For the $m$-th image, there exists an adjacency matrix $\mathbf{A}_m \in \mathbb{R}^{10000 \times 10000}$ and a node signal $\boldsymbol{x}_m \in \mathbb{R}^{10000}$. Based on the raw signal of each node, we apply two hybrid predefined filters to each image. Models are expected to learn these predefined filtering patterns. More details can be seen in Appendix D.2.1.

**Setup.** We compare PolyAttn with 5 baseline methods, including GCN (Kipf & Welling, 2017), GAT (Velickovic et al., 2018), GPRGNN Chien et al. (2021), BernNet (He et al., 2021), and Cheb-NetII (He et al., 2022). For PolyAttn, we employ both Monomial and Chebyshev bases, denoted as "PolyAttn (Mono)" and "PolyAttn (Cheb)," respectively. To ensure a fair comparison, all models are constrained to one single layer and have approximately 5k parameters. The learning rate is uniformly set to $0.001$, the training epochs to $50,000$, and the early stopping threshold to $400$ iterations. We employ two metrics to evaluate each method: the sum of squared errors and the $R^2$ score.

**Results.** As demonstrated in Table 2, PolyAttn outperforms other polynomial GNNs on all datasets. Compared to traditional polynomial GNNs, which employ unified coefficients for all nodes within a graph, PolyAttn uses tailored attention mechanisms for polynomial tokens to enable node-wise filtering. This design choice endows PolyAttn with greater expressive power. Further evidence of this capability is provided in Figures 2a and 2b. In this figure, filters learned for all nodes are divided into one of two clusters using the $k$-means (Jain & Dubes, 1988) algorithm, and the representative filter (centroid) for each cluster is plotted. PolyAttn is shown to successfully derive individual filter patterns without requiring prior knowledge of predefined filters. This underscores PolyAttn's ability to learn graph filters for each node adaptively.

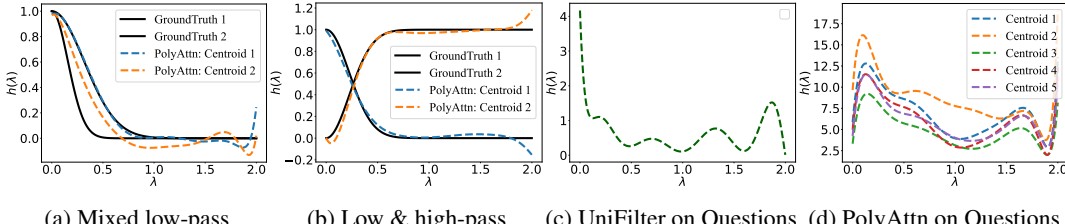

(a) Mixed low-pass      (b) Low & high-pass      (c) UniFilter on Questions    (d) PolyAttn on Questions

Figure 2: Left: Learned Filters of PolyAttn on Synthetic Datasets. Right: Learned Filters of UniFilter and PolyAttn on the real-world Questions Dataset.

### 4.1.2 PERFORMANCE ON REAL-WORLD DATASETS

**Setup.** We choose four real-world datasets to evaluate the efficacy of PolyAttn as a node-wise filter, including two homophilic graphs (PubMed (Namata et al., 2012) and CS (Shchur et al., 2018)) and two heterophilic graphs (Roman-empire and Questions (Platonov et al., 2023)). Dataset details and other settings are listed in Appendix D.1. As baseline models, we employ Monomial-based filters with uniform coefficients, denoted as "UniFilter (Mono)," and Chebyshev-based filters, denoted as "UniFilter (Cheb)." Correspondingly, we use node-wise filters based on PolyAttn, denoted as "PolyAttn (Mono)" and "PolyAttn (Cheb)." All models are configured with a single filtering layer to ensure a fair comparison. More details are listed in Appendix D.2.2.

---

[1] https://ww2.mathworks.cn/products/image.html

Table 3: Performance of PolyAttn on Real-world Datasets.

|  | CS | Pubmed | Roman-empire | Questions |
|---|---|---|---|---|
| UniFilter (Mono) | 95.32±0.24 | 89.61±0.44 | 73.44±0.80 | 73.19±1.52 |
| PolyAttn (Mono) | 95.99±0.07 | 90.85±0.31 | 74.17±0.59 | 76.83±0.79 |
| Improvement (%) | 0.70 | 1.38 | 0.99 | 4.96 |
| UniFilter (Cheb) | 96.17±0.10 | 88.65±0.35 | 72.81±0.73 | 74.55±0.78 |
| PolyAttn (Cheb) | 96.03±0.15 | 89.85±0.46 | 74.03±0.45 | 75.90±0.72 |
| Improvement (%) | -0.15 | 1.35 | 1.68 | 1.81 |

**Results.** Table 3 shows the mean accuracies with a 95% confidence interval over 10 runs. We observe that PolyAttn performs better on both homophilic and heterophilic graphs, with especially notable improvements on the latter one, which suggests the benefits of its node-wise filtering ability. Further insights are illustrated in Figures 2c and 2d, which show the learned filters by "UniFilter(Cheb)" and "PolyAttn(Cheb)" on Questions. The node-wise filters learned by PolyAttn are categorized into one of five clusters using the k-means algorithm (Jain & Dubes, 1988). Interestingly, we observe that PolyAttn learns various filters on different graph nodes. Given that PolyAttn improvements UniFilter by at least 1.8% on Questions, it may suggest that these node-wise filters are necessary. More illustrations on other datasets are available in Appendix E.

## 4.2 POLYFORMER EXPERIMENTS

### 4.2.1 NODE CLASSIFICATION

**Setup.** We employ datasets including four homophilic datasets (Sen et al., 2008; Shchur et al., 2018) and four heterophilic datasets (Platonov et al., 2023). Detailed characteristics and splits of these datasets are provided in Appendix D.1. As for baselines, we select several recent state-of-the-art spectral GNNs, including GPRGNN (Chien et al., 2021), BernNet (He et al., 2021), and ChebNetII (He et al., 2022). Additionally, Our comparison includes competitive graph Transformer models NAGphormer (Chen et al., 2023) and Specformer (Bo et al., 2023) to further underscore the performance of our approach. We report the mean accuracy with a 95% confidence interval over 10 runs. More details are displayed in Appendix D.3.1.

Table 4: Performance of PolyFormer on Node Classification. "OOM" means "out of memory," and "*" indicates the use of truncated eigenvalues and eigenvectors as suggested by Bo et al. (2023).

|  | Homophilic | | | | Heterophilic | | | |
|---|---|---|---|---|---|---|---|---|
|  | Citeseer | CS | Pubmed | Physics | Minesweeper | Tolokers | Roman-empire | Questions |
| MLP | 78.74±0.64 | 95.53±0.13 | 87.06±0.35 | 97.10±0.71 | 50.97±0.54 | 74.12±0.48 | 66.64±0.32 | 71.87±0.41 |
| GCN | 80.16±1.09 | 94.95±0.17 | 87.34±0.37 | 97.74±0.35 | 72.23±0.56 | 77.22±0.73 | 53.45±0.27 | 76.28±0.64 |
| GAT | 80.67±1.05 | 93.93±0.26 | 86.55±0.36 | 97.82±0.28 | 81.39±1.69 | 77.87±1.00 | 51.51±0.86 | 74.94±0.56 |
| GPRGNN | 80.61±0.75 | 95.26±0.15 | 91.00±0.34 | 97.74±0.35 | 90.10±0.34 | 77.25±0.61 | 74.08±0.54 | 74.36±0.67 |
| BernNet | 79.63±0.78 | 95.42±0.29 | 90.56±0.40 | 97.64±0.38 | 77.93±0.59 | 76.83±0.53 | 72.70±0.30 | 74.25±0.73 |
| ChebNetII | 80.25±0.65 | 96.33±0.12 | 90.60±0.17 | 97.25±0.78 | 83.64±0.40 | 79.23±0.43 | 74.64±0.39 | 74.41±0.58 |
| Transformer | 78.70±0.59 | OOM | 89.10±0.43 | OOM | 50.29±1.09 | 74.24±0.58 | 65.29±0.47 | OOM |
| Specformer | 81.69±0.78 | 96.07±0.10 | 89.94±0.33 | 97.70±0.60* | 89.93±0.41 | 80.42±0.55 | 69.94±0.34 | 76.49±0.58* |
| NAGphormer | 79.77±0.81 | 95.89±0.13 | 89.65±0.45 | 97.23±0.23 | 88.06±0.43 | 81.57±0.44 | 74.45±0.48 | 75.13±0.70 |
| PolyFormer (Mono) | 82.37±0.65 | 96.49±0.09 | 91.01±0.41 | 98.42±0.16 | 90.69±0.38 | 84.00±0.45 | 78.89±0.39 | 77.46±0.65 |
| PolyFormer (Cheb) | 81.80±0.76 | 96.49±0.17 | 90.68±0.31 | 98.08±0.27 | 91.90±0.35 | 83.88±0.33 | 80.27±0.39 | 77.26±0.50 |

**Results.** As shown in Table 4, our model consistently outperforms most baseline models, especially excelling on heterophilic datasets. Notably, when compared with spectral GNNs like BernNet (He et al., 2021) and ChebNetII (He et al., 2022), which utilize sophisticated polynomial bases such as Bernstein or advanced techniques like Chebyshev Interpolation, our model showcases superior performance. Such results suggest that the introduction of node-wise coefficients significantly boosts our model's expressive power. Furthermore, our model maintains competitive performance against transformer-based approaches. This observation indicates that focusing on information within a limited scope, i.e., a truncation of polynomial basis, appears to provide the necessary expressiveness for achieving competitive results. Conversely, taking all node pairs into account may introduce redundant noise that diminishes the model's performance.

Table 5: Performance of PolyFormer for Node Classification on Large-Scale Datasets. "-" means "out of memory" or failing to complete preprocessing within 24 hours.

|  | Twitch-Gamers | ogbn-arxiv | Pokec | ogbn-papers100M |
| --- | --- | --- | --- | --- |
| MLP | 60.92±0.07 | 55.50±0.23 | 62.37±0.02 | 47.24±0.31 |
| GCN | 62.18±0.26 | 71.74±0.29 | 75.45±0.17 | - |
| ChebNet | 62.31±0.37 | 71.12±0.22 |  | - |
| GPR-GNN | 62.59±0.38 | 71.78±0.18 | 80.74±0.22 | 65.89±0.35 |
| Specformer | 64.22±0.04 | 72.37±0.18 | - | - |
| NAGphormer | 64.38±0.04 | 71.04±0.94 | - | - |
| NodeFormer | 61.12±0.05 | 60.02±0.52 | 70.48±0.45 | - |
| PolyFormer | **64.79±0.10** | **72.42±0.19** | **82.29±0.14** | **67.11±0.20** |

### 4.2.2 NODE CLASSIFICATIONS ON LARGE-SCALE DATASETS

**Setup.** We perform node classification tasks on two expansive citation networks: ogbn-arxiv and ogbn-papers100M (Hu et al., 2020), in addition to two large-scale heterophilic graphs: Twitch-Gamers and Pokec, sourced from (Lim et al., 2021) to demonstrating the scalability of our model. More information is provided in Appendix D.1. We select common GNN models, including (Kipf & Welling, 2017; Chien et al., 2021; Defferrard et al., 2016). For graph Transformer models, we use expressive Specformer (Bo et al., 2023), and two scalable baseline Nodeformer (Wu et al., 2022) and NAGphormer (Chen et al., 2023). More details are available in Appendix D.3.2.

**Results.** Table 5 shows the mean accuracies over multiple runs. Due to our efficient node tokenization techniques, PolyFormer exhibits great scalability up to the graph ogbn-papers100M, which has over **100 million nodes**. In contrast, models such as NAGphormer (Chen et al., 2023) and Specformer (Bo et al., 2023) rely on Laplacian eigenvectors or eigenvalues, which constrains their scalability. Moreover, by leveraging expressive PolyAttn, our model outperforms all baselines.

### 4.3 COMPLEXITY COMPARISON

In this subsection, we evaluate PolyFormer in comparison to other graph Transformer models concerning time and GPU memory consumption. More details are available in Appendix D.4.

**Results.** Table 6 illustrates that our PolyFormer is significantly faster and more memory-efficient compared to other Transformer-based models. Firstly, in comparison to the addition of position encoding using Laplacian eigenvectors or eigenvalues, the preprocessing of polynomial tokens is substantially quicker. Secondly, our proposed PolyAttn exhibits lower complexity compared to methods that implement attention on node pairs, resulting in minimal runtime and GPU usage. Both of these are theoretically analyzed in Section 3.3.1. In conclusion, our proposed model performs well in both running efficiency and memory consumption.

Table 6: Model Performance Comparison on Roman-empire.

| Model | Prep. Time (s) | Train. (ms/epoch) | Infer. (ms/epoch) | Max GPU Mem. (MB) |
| --- | --- | --- | --- | --- |
| Transformer | 0 | 1676.35 | 786.27 | 8640.65 |
| NAGphormer | 45.83 | 351.83 | 708.21 | 1277.15 |
| Specformer | 1270.88 | 992.24 | 1945.89 | 10714.23 |
| PolyFormer | 0.89 | 300.85 | 291.53 | 1207.06 |

## 5 CONCLUSION

In this study, we introduce a novel formulation of node tokens that leverages polynomial bases to efficiently capture spectral domain information. Further, we propose PolyAttn, which functions as a node-wise filter. This approach outperforms traditional node-unified filters in terms of expressiveness while simultaneously avoiding computational overhead. Building on polynomial tokens and PolyAttn, we present PolyFormer, a scalable graph Transformer designed specifically for node-level tasks. PolyFormer strikes a balance between expressive power and computational efficiency. Extensive empirical evaluations corroborate the superior performance, scalability, and efficiency of PolyFormer. A promising future direction is to improve PolyFormer with more advanced polynomial approximation and graph spectral techniques.

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

# A    NOTATIONS

Table 7: Summary of notations in this paper.

| Notation | Description |
|---|---|
| $G = (V, E)$ | A graph where $V$ is the set of nodes and $E$ is the set of edges. |
| $N$ | Total number of nodes in the graph. |
| $\mathbf{A}(\hat{\mathbf{A}})$ | The adjacency matrix of the graph and its normalized version. |
| $\hat{\mathbf{L}}$ | Normalized Laplacian of the graph. |
| $\mathbf{P}$ | Refers to either $\hat{\mathbf{A}}$ or $\hat{\mathbf{L}}$. |
| $\mathbf{X} \in \mathbb{R}^{N \times d}$ | Original graph signal matrix or node feature matrix. |
| $\mathbf{Z} \in \mathbb{R}^{N \times d}$ or $\mathbf{Z} \in \mathbb{R}^{N \times c}$ | Filtered signal or representation of nodes. |
| $\{g_k(\cdot)\}_{k=0}^K$ | Series polynomial basis of truncated order $K$. |
| $\{\alpha_k\}_{k=0}^K$ | Polynomial coefficients for all nodes, i.e.$\mathbf{Z} \approx \sum_{k=0}^K \alpha_k g_k(\mathbf{P})\mathbf{X}$. |
| $\{\alpha_k^{(i)}\}_{k=0}^K$ | Polynomial coefficients of nodes $v_i$, i.e.$\mathbf{Z}_{i,:} \approx \sum_{k=0}^K \alpha_k^{(i)} (g_k(\mathbf{P})\mathbf{X})_{i,:}$. |
| $\{\alpha_{(p,q)k}\}_{k=0}^K$ | Coefficients on channel (p,q),i.e.$\mathbf{Z}_{:,p:q} \approx \sum_{k=0}^K \alpha_{(p,q)k} (g_k(\mathbf{P})\mathbf{X})_{:,p:q}$. |
| $\boldsymbol{h}_k^{(i)} \in \mathbb{R}^d$ | Polynomial token of order $k$ for node $v_i$. |
| $\mathbf{H}_k \in \mathbb{R}^{N \times d}$ | Matrix contains order-$k$ polynomial tokens for all nodes. |
| $\mathbf{H}^{(i)} \in \mathbb{R}^{(K+1) \times d}$ | Token matrix for node $v_i$. |
| $\boldsymbol{\beta} \in \mathbb{R}^{(K+1)}$ | Attention bias vector shared across all nodes. |
| $\mathbf{B} \in \mathbb{R}^{(K+1) \times (K+1)}$ | Attention bias matrix, where each entry $\mathbf{B}_{ij}$ equals $\beta_j$. |
| $\mathbf{Q}, \mathbf{K}, \mathbf{V} \in \mathbb{R}^{(K+1) \times d}$ | The query, key, and value matrices, respectively. |
| $\mathbf{S} \in \mathbb{R}^{(K+1) \times (K+1)}$ | The attention score matrix. |

# B    PROOF

## B.1    PROOF OF THE THEOREM

Here we provide the detailed proof for Theorem 3.1.

*Proof.* For a node $v_i$ in the graph, the corresponding token matrix is given by $\mathbf{H}^{(i)} = [\boldsymbol{h}_0^{(i)}, \cdots, \boldsymbol{h}_K^{(i)}]^\top \in \mathbb{R}^{(K+1) \times d}$. When processed by the order-wise MLP, each row $\mathbf{H}_{j,:}^{(i)}$ is updated as $\mathbf{H}_{j,:}^{(i)} = \text{MLP}_j(\mathbf{H}_{j,:}^{(i)})$. Subsequently, the query matrix $\mathbf{Q}$ and the key matrix $\mathbf{K}$ are calculated as $\mathbf{Q} = \mathbf{H}^{(i)} \mathbf{W}_Q$ and $\mathbf{K} = \mathbf{H}^{(i)} \mathbf{W}_K$, respectively. The attention matrix $\mathbf{A}_{attn} \in \mathbb{R}^{(K+1) \times (K+1)}$ is then formulated as follows:

$$\mathbf{A}_{attn} = \begin{bmatrix} a_{00} & a_{01} & \cdots & a_{0K} \\ a_{10} & a_{11} & \cdots & a_{1K} \\ \vdots & \vdots & \ddots & \vdots \\ a_{K0} & a_{K1} & \cdots & a_{KK} \end{bmatrix}, \tag{10}$$

where $a_{ij} = (\mathbf{Q}\mathbf{K}^\top)_{ij}$.

Taking the activation function $\sigma$ and the attention bias matirx $\mathbf{B}$ into account, the corresponding attention score matrix $\mathbf{S} = \mathbf{A}_{attn} \odot \mathbf{B} \in \mathbb{R}^{(K+1) \times (K+1)}$, where $\mathbf{B}_{ij} = \beta_j, j \in \{0, \cdots, K\}$.

According to $\mathbf{H}'^{(i)} = \mathbf{S}\mathbf{V}$ and $\mathbf{V} = \mathbf{H}^{(i)}$, we have:

$$\mathbf{H}'^{(i)} = \mathbf{S}\mathbf{H}^{(i)} = \begin{bmatrix} s_{00} & s_{01} & \cdots & s_{0K} \\ s_{10} & s_{11} & \cdots & s_{1K} \\ \vdots & \vdots & \ddots & \vdots \\ s_{K0} & s_{K1} & \cdots & s_{KK} \end{bmatrix} \left[ \boldsymbol{h}_0^{(i)}, \cdots, \boldsymbol{h}_K^{(i)} \right]^\top = \left[ \sum_{k=0}^K s_{0k} \boldsymbol{h}_k^{(i)}, \cdots, \sum_{k=0}^K s_{Kk} \boldsymbol{h}_k^{(i)} \right]^\top,$$
$$\tag{11}$$

where $\mathbf{H}'^{(i)} \in \mathbb{R}^{(K+1) \times d}$. As representation of node $v_i$ is calculated by $\mathbf{Z}_{i,:} = \sum_{k=0}^{K} \mathbf{H}'^{(i)}_{k,:}$, we have:

$$
\begin{aligned}
\mathbf{Z}_{i,:} &= \sum_{k=0}^{K} \mathbf{H}'^{(i)}_{k,:} \\
&= \sum_{k=0}^{K} s_{0k} \boldsymbol{h}_k^{(i)} + \cdots + \sum_{k=0}^{K} s_{Kk} \boldsymbol{h}_k^{(i)} \\
&= \sum_{k=0}^{K} s_{k0} \boldsymbol{h}_0^{(i)} + \cdots + \sum_{k=0}^{K} s_{kK} \boldsymbol{h}_K^{(i)} \\
&= \alpha_0^{(i)} \boldsymbol{h}_0^{(i)} + \cdots + \alpha_K^{(i)} \boldsymbol{h}_K^{(i)} \\
&= \sum_{k=0}^{K} \alpha_k^{(i)} \boldsymbol{h}_k^{(i)} \\
&= \sum_{k=0}^{K} \alpha_k^{(i)} \left( g_k \left( \mathbf{P} \right) \mathbf{X} \right)_{i,:}.
\end{aligned}
\tag{12}
$$

Here, $\alpha_j^{(i)}$ denotes

$$
\alpha_j^{(i)} = \sum_{k=0}^{K} s_{kj}, \quad j \in \{0, \cdots, K\}
$$

and is computed based on the node's token matrix $\mathbf{H}^{(i)}$. This value serves as a **node-wise** weight for the polynomial filter and is determined by both the node features and the topology information of the node $v_i$. Consequently, the described PolyAttn mechanism functions as a node-wise filter. $\square$

### B.2 PROOF OF THE PROPOSITIONS

In the following, we present a proof for Proposition 3.1.

*Proof.* For node $v_i$, the multi-head PolyAttn mechanism employs the sub-channel of the token matrix $\mathbf{H}^{(i)}_{:,jd_h:(j+1)d_h-1}$ for head $j$, where $j \in \{0, \ldots, h-1\}$.

According to Theorem 3.1, there exists a set of node-wise coefficients for node $v_i$, denoted by $\alpha^{(i)}_{(jd_h,(j+1)d_h-1)k}$, with $k \in \{0, \ldots, K\}$. These coefficients are computed based on the corresponding sub-channel of the token matrix $\mathbf{H}^{(i)}_{:,jd_h:(j+1)d_h-1}$. The contribution of head $j$ to the node representation $\mathbf{Z}_{i,:}$ can then be formally expressed as:

$$
\mathbf{Z}_{i,jd_h:(j+1)d_h-1} = \sum_{k=0}^{K} \alpha^{(i)}_{(jd_h,(j+1)d_h-1)k} \left( g_k(\mathbf{P})\mathbf{X} \right)_{i,jd_h:(j+1)d_h-1}.
\tag{13}
$$

By concatenating the contributions from all heads, we obtain the complete node representation for node $v_i$. Throughout this procedure, the multi-head PolyAttn mechanism performs a filtering operation on each channel group separately. $\square$

Below, we deliver a detailed proof for Proposition 3.2.

*Proof.* According to Proof B.1, when the PolyAttn functions as a node-wise filter for node $v_i$, we have:

$$
\mathbf{Z}_{i,:} = \sum_{k=0}^{K} \mathbf{H}'^{(i)}_{k,:} = \sum_{k=0}^{K} \alpha_k^{(i)} \left( g_k \left( \mathbf{P} \right) \mathbf{X} \right)_{i,:},
$$

where

$$\alpha_j^{(i)} = \sum_{k=0}^{K} s_{kj} = \sum_{k=0}^{K} \sigma(a_{kj})\beta_j, j \in \{0, \cdots, K\}.$$

If the softmax function is employed, then for any node $v_i$ in the graph, the value of $\sum_{k=0}^{K} \sigma(a_{kj})$ remains positive after the softmax operation. The sign of $\alpha_j^{(i)}$ is thus determined by the bias $\beta_j$. Since this bias is not node-specific, it implies that the coefficients of all nodes are constrained by the bias $\beta_j$, thereby limiting the expressive power of PolyAttn when acting as a node-wise filter. For instance, when all biases $\beta_j$ are positive, then

$$\alpha_j^{(i)} = \sum_{k=0}^{K} s_{kj} = \sum_{k=0}^{K} \sigma(a_{kj})\beta_j > 0,$$

PolyAttn with a Monomial basis can only serve as a low-pass filter for all nodes (Chien et al., 2021). In contrast, the activation function $\tanh(\cdot)$ allows the coefficient $\alpha_j^{(i)} = \sum_{k=0}^{K} s_{kj} = \sum_{k=0}^{K} \sigma(a_{kj})\beta_j$ to vary across nodes, enhancing the expressive power of PolyAttn. $\square$

## C IMPLEMENTATION DETAILS

**Multi-head PolyAttn.** Here we provide pseudocode for the multi-head PolyAttn mechanism as below:

---
**Algorithm 2:** Pseudocode for Multi-head PolyAttn

---
**Input:** Token matrix for node $v_i$: $\mathbf{H}^{(i)} = [\boldsymbol{h}_0^{(i)}, \cdots, \boldsymbol{h}_K^{(i)}]^\top \in \mathbb{R}^{(K+1) \times d}$
**Output:** New token matrix for node $v_i$: $\mathbf{H}'^{(i)} \in \mathbb{R}^{(K+1) \times d}$
**Learnable Parameters:** Projection matrix $\mathbf{W}_Q, \mathbf{W}_K \in \mathbb{R}^{d \times (d_h \times h)}$,
                          token-wise $\text{MLP}_j (j = 0, \cdots, K)$,
                          attention bias $\mathbf{B} \in \mathbb{R}^{(h \times (K+1)}$

1   Initialize $\mathbf{V}$ with $\mathbf{H}^{(i)}$
2   **for** $j = 0$ **to** $K$ **do**
3     $\quad \mathbf{H}_{j,:}^{(i)} \leftarrow \text{MLP}_j(\mathbf{H}_{j,:}^{(i)})$
4   $\mathbf{Q} \leftarrow \mathbf{H}^{(i)}\mathbf{W}_Q$ via projection matrix $\mathbf{W}_Q$; $\mathbf{K} \leftarrow \mathbf{H}^{(i)}\mathbf{W}_K$ via projection matrix $\mathbf{W}_K$
5   Reshape $\mathbf{Q}, \mathbf{K}$ into $h$ heads to get
     $\mathbf{Q}_{(m)} \in \mathbb{R}^{(K+1) \times d_h}, \mathbf{K}_{(m)} \in \mathbb{R}^{(K+1) \times d_h}, m \in \{0, \cdots, h-1\}$
6   **for** $m = 0$ **to** $h - 1$ **do**
7     $\quad \mathbf{S}_{(m)} \leftarrow \tanh(\mathbf{Q}_{(m)}\mathbf{K}_{(m)}^\top) \odot \mathbf{B}_{m,j}$
8     $\quad \mathbf{H}'^{(i)}_{(m)} \leftarrow \mathbf{S}_{(m)}\mathbf{V}_{:p,q}$, where $(p, q) = (d_h \times m, d_h \times (m+1) - 1)$
9   $\mathbf{H}'^{(i)} \leftarrow [\mathbf{H}'^{(i)}_{(0)} || \cdots || \mathbf{H}'^{(i)}_{(h-1)}] \in \mathbb{R}^{(K+1) \times d}$, where $[\cdot || \cdots || \cdot]$ means concatenating matrices
10   **return** $\mathbf{H}'^{(i)}$

---

**Attention Bias.** In implementation, we imposed constraints on the bias corresponding to each order of polynomial tokens. Specifically, for the learnable bias $\boldsymbol{\beta}$, the attention bias matrix $\mathbf{B} \in \mathbb{R}^{(K+1) \times (K+1)}$ is defined as $\mathbf{B}_{ij} = \frac{\beta_j}{(j+1)^r}$, where hyperparameter $r$ is the constraint factor.

**Order-wise MLP.** To enhance the expressive capacity of the order-wise MLP, we use the hyperparameter $m$ to increase the intermediate dimension of the order-wise MLP. Specifically, for an input dimension $d$ the intermediate dimension of the order-wise MLP is $m \times d$.

# D    Experimental Settings

## D.1    Dataset Description

### D.1.1    Dataset Statistics

Here we provide detailed characteristics of all datasets used in the experiments, as shown in Table 8 and Table 9.

Table 8: Statistics of Real-world Datasets.

| Dataset | Citeseer | CS | Pubmed | Physics | Minesweeper | Tolokers | Roman-empire | Questions |
|---|---|---|---|---|---|---|---|---|
| **Nodes** | 3,327 | 18,333 | 19,717 | 34,493 | 10,000 | 11,758 | 22,662 | 48,921 |
| **Edges** | 9,104 | 163,788 | 44,324 | 495,924 | 39,402 | 519,000 | 32,927 | 153,540 |
| **Features** | 3,703 | 6,805 | 500 | 8,415 | 7 | 10 | 300 | 301 |
| **Classes** | 6 | 15 | 3 | 5 | 2 | 2 | 18 | 2 |

Table 9: Statistics of Large-scale Datasets.

| Dataset | Twitch-Gamers | ogbn-arxiv | pokec | ogbn-papers100M |
|---|---|---|---|---|
| **Nodes** | 168,114 | 169,343 | 1,632,803 | 111,059,956 |
| **Edges** | 6,797,557 | 1,166,243 | 30,622,564 | 1,615,685,872 |
| **Features** | 7 | 128 | 65 | 128 |
| **Classes** | 2 | 40 | 2 | 172 |

### D.1.2    Dataset Splits

For homophilic datasets including Citeseer, CS, Pubmed, and Physics, we employ a random split: 60% for the training set, 20% for the validation set, and 20% for the test set, following the approach of (He et al., 2021).

For heterophilic graphs, namely Minesweeper, Tolokers, Roman-empire, and Questions, we adopt the given split: 50% for training, 25% for validation, and 25% for testing, as provided in (Platonov et al., 2023).

For large-scale datasets, we utilize the split from (Lim et al., 2021) for Twitch-gamers and pokec. Meanwhile, for ogbn-arxiv and ogbn-papers100M, we adhere to the given splits as presented in (Hu et al., 2020).

## D.2    PolyAttn Experiments

### D.2.1    Fitting Signals in Synthetic Datasets

Based on the raw signal of each node in a graph, we apply one of two predefined filters. For example, for nodes with signals $x_1 < 0.5$, we define a low-pass filter $h_1(\lambda) = \exp(-10\lambda^2)$, resulting in a filtered signal $z_1 = \mathbf{U}h_1(\mathbf{\Lambda})\mathbf{U}^\top x_1$. Conversely, for nodes with signals $x_1 \geq 0.5$, we implement a high-pass filter $h_2(\lambda) = 1 - \exp(-10\lambda^2)$, yielding the corresponding filtered signal $z_2 = \mathbf{U}h_2(\mathbf{\Lambda})\mathbf{U}^\top x_2$. For eigenvalues $\lambda \in [0, 2]$, the predefined filters $h_1(\lambda)$ and $h_2(\lambda)$ are presented in Table 10. Given the original graph signals $x_1, x_2$ and the filtered graph signals $z_1, z_2$, Models are expected to learn these filtering patterns.

In this experiment, every model uses a truncated order of $K = 10$ within one layer. Additionally, We employed one head for PolyAttn. All models have total parameters of approximately $50,000$, achieved by using an adaptive hidden dimension.

### D.2.2    Performance on Real-World Datasets

To ensure a fair comparison, the truncated order $K$ is set to 10 and the layer to be 1 for both the node-unified filter and PolyAttn. We also set the number of heads for PolyAttn as 1.

| Filters | $h_1(\lambda)$ | $h_2(\lambda)$ |
|---|---|---|
| Mixed low-pass | $h(\lambda) = e^{-5\lambda^2}$ | $h(\lambda) = e^{-20\lambda^2}$ |
| Mixed high-pass | $h(\lambda) = 1 - e^{-5\lambda^2}$ | $h(\lambda) = 1 - e^{-20\lambda^2}$ |
| Mixed band-pass | $h(\lambda) = e^{-5(\lambda-1)^2}$ | $h(\lambda) = e^{-20(\lambda-1)^2}$ |
| Mixed rejection-pass | $h(\lambda) = 1 - e^{-5(\lambda-1)^2}$ | $h(\lambda) = 1 - e^{-20(\lambda-1)^2}$ |
| Low & high-pass | $h(\lambda) = 1 - e^{-10\lambda^2}$ | $h(\lambda) = e^{-10\lambda^2}$ |
| Band & rejection-pass | $h(\lambda) = 1 - e^{-10(\lambda-1)^2}$ | $h(\lambda) = e^{-10(\lambda-1)^2}$ |

Table 10: The predefined filters on graph signals.

Hyperparameters, including hidden dimensions, learning rates, and weight-decay rates, are fine-tuned through 200 rounds of Optuna (Akiba et al., 2019) hyperparameter search. The best configuration is chosen based on its performance on the validation set. The final outcomes are the averages of 10 evaluations on the test set with a 95% confidence interval using the optimal parameters.

The Optuna search space consists of 100 trials, with the searching space provided below:

- Hidden dimension: $\{16, 32, 64, 128, 256\}$;
- Learning rates: $\{5\text{e-}5, 2\text{e-}4, 1\text{e-}3, 1\text{e-}2\}$;
- Weight decays: $\{0.0, 1\text{e-}5, 1\text{e-}4, 5\text{e-}4, 1\text{e-}3\}$;
- Dropout rates: $\{0.0, 0.1, 0.2, \ldots, 0.9\}$;

There is one extra hyperparameter for PolyAttn:

- Multiplying factor $m$ for order-wise MLP: $\{1.0, 2.0, 0.5\}$.

### D.3 POLYFORMER EXPERIMENTS

#### D.3.1 NODE CLASSIFICATIONS

We train all models with the Adam optimizer (Kingma & Ba, 2015). Early stopping is employed with a patience of 200 epochs out of a total of 2000 epochs. The mean test accuracy, along with a 95% confidence interval, is reported based on 10 runs.

Hyperparameter selection is carried out on the validation sets. To expedite the hyperparameter selection process, we utilize Optuna (Akiba et al., 2019), performing a maximum of 400 complete trials within the following hyperparameter ranges:

- Truncated order $K$ of polynomial tokens: $\{2, 4, 6, 8, 10, 12, 14\}$;
- Number of layers: $\{1, 2, 3, 4\}$;
- Number of heads: $\{1, 2, 4, 8, 16\}$;
- Hidden dimension: $\{16, 32, 64, 128, 256\}$;
- Hidden size for FFN: $\{32, 64, 128, 256, 512\}$;
- Learning rates: $\{0.00005, 0.0001, 0.0005, 0.001, 0.005, 0.01\}$;
- Weight decays: $\{0.0, 1\text{e-}8, 1\text{e-}7, 1\text{e-}6, 1\text{e-}5, 1\text{e-}4, 1\text{e-}3\}$;
- Dropout rates: $\{0.0, 0.1, 0.2, \ldots, 0.9\}$;
- Constraint factor $r$: $\{1.0, 1.2, 1.4, 1.6, 1.8, 2.0\}$;
- Multiplying factor $m$ for order-wise MLP : $\{1.0, 2.0, 0.5\}$.

As demonstrated in Tables 11 and 12, the primary hyperparameters utilized in our model for node classification tasks are presented. Further details will be provided in the accompanying code release.

#### D.3.2 NODE CLASSIFICATIONS ON LARGE-SCALE DATASETS

The reported results for GNNs are sourced from He et al. (2022), whereas those for the Graph Transformer are derived from the recommended hyperparameters or hyperparameter searching. The

| Dataset | K | layers | heads | lr | wd | hidden | d_ffn | dropout | r | m |
|---------|---|--------|-------|----|----|--------|-------|---------|---|---|
| Citeseer | 6 | 1 | 4 | 5e-4 | 1e-3 | 256 | 128 | 0.9 | 2.0 | 2.0 |
| CS | 2 | 1 | 8 | 1e-3 | 1e-07 | 128 | 128 | 0.0 | 1.0 | 1.0 |
| Pubmed | 2 | 2 | 8 | 5e-3 | 1e-3 | 256 | 32 | 0.5 | 1.6 | 2.0 |
| physics | 4 | 1 | 2 | 1e-3 | 1e-05 | 128 | 256 | 0.9 | 1.2 | 0.5 |
| Minesweeper | 10 | 4 | 8 | 0.01 | 1e-05 | 16 | 32 | 0.2 | 1.6 | 2.0 |
| Tolokers | 10 | 1 | 16 | 5e-4 | 1e-08 | 64 | 128 | 0.2 | 1.0 | 1.0 |
| Roman-empire | 14 | 3 | 16 | 1e-4 | 1e-3 | 256 | 64 | 0.5 | 1.0 | 2.0 |
| Questions | 12 | 1 | 4 | 5e-4 | 1e-3 | 128 | 256 | 0.2 | 1.0 | 1.0 |

Table 11: Hyperparameters for PolyFormer (Mono).

| Dataset | K | layers | heads | lr | wd | hidden | d_ffn | dropout | r | m |
|---------|---|--------|-------|----|----|--------|-------|---------|---|---|
| Citeseer | 6 | 1 | 4 | 5e-4 | 1e-3 | 256 | 128 | 0.9 | 2.0 | 2.0 |
| CS | 2 | 1 | 8 | 1e-3 | 1e-07 | 128 | 128 | 0.0 | 1.0 | 1.0 |
| Pubmed | 2 | 2 | 8 | 5e-3 | 1e-3 | 256 | 32 | 0.5 | 1.6 | 2.0 |
| physics | 4 | 1 | 2 | 1e-3 | 1e-05 | 128 | 256 | 0.9 | 1.2 | 0.5 |
| Minesweeper | 10 | 4 | 8 | 0.01 | 1e-05 | 16 | 32 | 0.2 | 1.6 | 2.0 |
| Tolokers | 10 | 1 | 16 | 5e-4 | 1e-08 | 64 | 128 | 0.2 | 1.0 | 1.0 |
| Roman-empire | 14 | 3 | 16 | 1e-4 | 1e-3 | 256 | 64 | 0.5 | 1.0 | 2.0 |
| Questions | 12 | 1 | 4 | 5e-4 | 1e-3 | 128 | 256 | 0.2 | 1.0 | 1.0 |

Table 12: Hyperparameters for PolyFormer (Cheb).

mean test accuracy, accompanied by a 95% confidence interval, is reported based on either 5 or 10 runs.

We utilize the Adam optimizer (Kingma & Ba, 2015) to train our models. Early stopping is implemented with patience at 250 epochs within an overall training span of 2000 epochs. The hyperparameter space used for experiments on large-scale datasets is enumerated below:

- Truncated order $K$ of polynomial tokens: $\{4, 8, 10\}$;
- Number of layers: $\{1, 2\}$;
- Number of heads: $\{1, 4, 8\}$;
- Hidden dimension: $\{128, 512\}$;
- Hidden size for FFN: $\{512, 1024\}$;
- Learning rates: $\{0.00005, 0.0002, 0.01\}$;
- Weight decays: $\{0.0, 0.00005, 0.0005, 0.001\}$;
- Dropout rates: $\{0.0, 0.25, 0.4, 0.6, 0.8\}$;
- Constraint factor $r$: $\{1.0, 2.0\}$;
- Multiplying factor $m$ for order-wise MLP: $\{0.5, 1.0\}$;
- Batch size: $\{10000, 20000, 50000\}$.

## D.4 COMPLEXITY COMPARISON

In the comparison experiment, we set the number of layers and the number of heads for all models to 1. To ensure a fair comparison, we kept the total number of parameters at approximately $50,000$, adjusting the hidden size as needed. Specifically, for NAGphormer, we set the position encoding dimension to 15, as suggested by Chen et al. (2023). For Specformer, we use all eigenvalues and eigenvectors, as Bo et al. (2023) recommends. For our proposed PolyFormer, we set the truncated order $K$ to 10, consistent with the number of hops for NAGphormer. Other hyperparameters, such as learning rates and weight decays, are kept consistent across all models.

# E  FILTERS OF POLYATTN

In this section, we present the filters learned by PolyAttn on real-world datasets, as implemented in Section 4.1.2.

As shown below, PolyAttn can learn node-wise filters, which provides it with greater expressive power compared to node-unified filters. Additionally, there is a noticeable increase in the variation of filters learned by PolyAttn for heterophilic datasets. In contrast, for homophilic graphs, PolyAttn tends to derive similar or identical filters across different nodes. These observations suggest that PolyAttn has the capability to adaptively learn filters based on the specific characteristics of each graph.

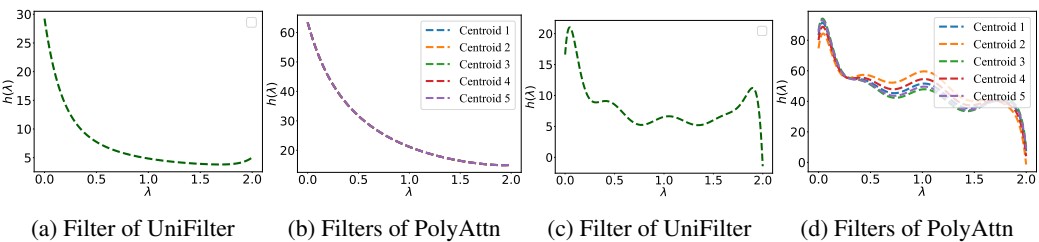

(a) Filter of UniFilter     (b) Filters of PolyAttn     (c) Filter of UniFilter     (d) Filters of PolyAttn

Figure 3: Filters Learned by UniFilter and PolyAttn on **PubMed** (*Monomial basis*, left and *Chebyshev basis*, right).

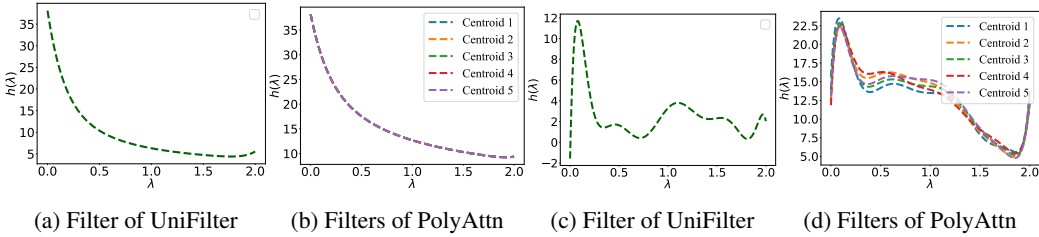

(a) Filter of UniFilter     (b) Filters of PolyAttn     (c) Filter of UniFilter     (d) Filters of PolyAttn

Figure 4: Filters Learned by UniFilter and PolyAttn on **CS** (*Monomial basis*, left and *Chebyshev basis*, right).

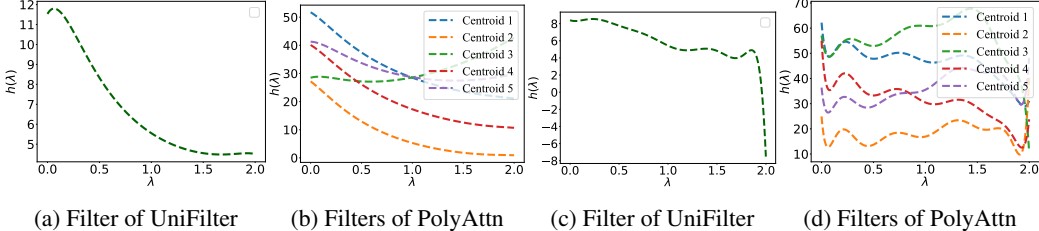

(a) Filter of UniFilter     (b) Filters of PolyAttn     (c) Filter of UniFilter     (d) Filters of PolyAttn

Figure 5: Filters Learned by UniFilter and PolyAttn on **Roman-empire** (*Monomial basis*, left and *Chebyshev basis*, right).

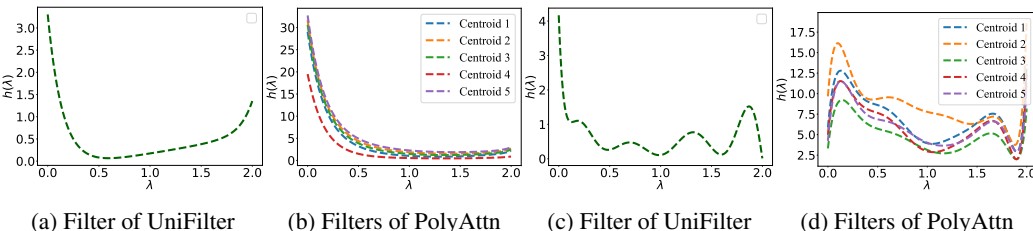

(a) Filter of UniFilter      (b) Filters of PolyAttn      (c) Filter of UniFilter      (d) Filters of PolyAttn

Figure 6: Filters Learned by UniFilter and PolyAttn on **Questions** (*Monomial basis*, left and *Chebyshev basis*, right).

