# OpenReview forum: "PolyFormer: Scalable Graph Transformer via Polynomial Attention"
_ICLR.cc/2024/Conference — Submitted to ICLR 2024_

### Official Review · Reviewer_13YF · 2023-10-23

**Soundness:** 3 good
**Presentation:** 3 good
**Contribution:** 3 good
**Rating:** 6
**Confidence:** 3

**Summary:**

This paper proposes a graph transformer architecture that uses intra-node attention from polynomial bases computed from said nodes. The method, PolyFormer, achieves good empirical performance on a variety of artifical and real-world tasks. The method is computationally cheaper than existing graph transformer approaches, which also aids scalability.

**Strengths:**

- The method achieves good empirical performance on a wide variety of artifical and real-world graph datasets.
- The method is computationally cheaper than calculating inter-node attention (O(L^2) instead of O(N^2)) but does not appear to suffer vs methods that do inter-node attention.
- The polynomial basis decomposition is interesting. Specifically, the monomial basis can be interpreted as the number of hops away from a node and still gets reasonable performance

**Weaknesses:**

- My understanding is that this method uses intra-node attention, where attention is calculated for a set of tokens associated with *each node* and not between nodes. The only cross-node information is from the tokens when they are constructed with A and L. This does not seem like a very efficient way of passing information between nodes, as the cross-node information is hardcoded in the tokens.
- The experiments do not show a large improvement over baseline methods. For example, in Table 2, the ChebNetII baseline performs very well and is not that far off PolyAttn (Cheb). In fact, it outperforms PolyAttn (Mono) on some tasks. This is a bit disappointing since PolyAttn (Mono) is touted as being interpretable, but PolyAttn (Cheb) is the method that outperforms baselines.
- I may have missed this, but are there any experiments showing how PolyAttn scales to more than one PolyAttn layer?

**Questions:**

See above.

---

> ### Author Response · Authors · 2023-11-15
> **Reply to Reviewer 13YF (1)**
>
> We are delighted to receive your positive feedback! Here are our responses and perspectives regarding the weaknesses you mentioned.
>
> **W1:** My understanding is that this method uses intra-node attention, where attention is calculated for a set of tokens associated with each node and not between nodes. The only cross-node information is from the tokens when they are constructed with A and L. This does not seem like a very efficient way of passing information between nodes, as the cross-node information is hardcoded in the tokens.
>
> **A1:** Our method employs intra-node attention. This choice is based on the following considerations:
>
> **Scalability**: In computer vision image classification, Transformer-based models usually calculate attention between patches of an image, rather than between entire images. Similarly, in natural language processing sentence classification, these models typically focus on attention between nodes within a sentence. This approach facilitates training and inference on mini-batches, thus avoiding scalability issues when dealing with a large number of images or sentences. Inspired by this, we design polynomial tokens for each node and compute attention between these tokens within a node for node classification tasks, thereby enhancing the model's scalability. Conversely, cross-node attention entails considerable computational costs, resulting in a complexity of $O(n^2d)$, where $n$ is the number of nodes, and $d$ is the dimensionality of the node representations. Such complexity is impractical for large-scale graph node classification tasks. Even with efficient attention mechanisms like [1], the complexity remains at least $O(nd^2)$, which is prohibitive for large-scale graphs with, for example, 100 million nodes. In short, intra-node attention enables Graph Transformer models with great scalability.
>
> **Model Performance**: While cross-node information can provide flexibility in learning node-pair relationships for graph Transformers, this flexibility can sometimes detract from practical performance, as noted in [2]. To incorporate structural information and boost performance, NAGphormer and [3, 4] have introduced various positional encodings. However, these additions incur extra costs, limiting scalability to large graph datasets and potentially failing to achieve ideal model performance. Although cross-node attention seems more efficient in transmitting information between nodes, most real-world graph data resemble small-world networks. In such networks, any node can be reached from any other node within a limited number of steps. Therefore, even with a truncated order of polynomial $h_k(\mathbf{A}, \mathbf{L})$, the polynomial tokens contain sufficient information for intra-node attention. Besides, our intra-node attention mechanism, PolyAttn, is theoretically equivalent to a node-wise polynomial filter in the spectral domain, as demonstrated in Theorem 3.1:
> $$
> \mathbf{Z}\_{i,:} = \sum\_{k=0}^K \mathbf{H}'^{(i)}\_{k,:} = \sum\_{k=0}^K \alpha\_k^{(i)}(g\_k(\mathbf{P})\mathbf{X})\_{i,:} = \left(\mathbf{U}h^{(i)}(\mathbf{\Lambda})\mathbf{U}^\top \mathbf{X}\right)_{i,:}
> $$
> This kind of filter [5, 6, 7, 8, 9], which hardcode node information via polynomial, perform well in node classification tasks and outperforms some Graph Transformer models which employ cross-node attention. In fact, our node classification experiments (Tables 4 and 5) demonstrate that our model using hardcore encoding delivers outstanding results in most cases as well.
>
> In summary, our intra-node attention mechanism, termed "PolyAttn", is designed considering both scalability and model performance.
>
> **W2:** The experiments do not show a large improvement over baseline methods. For example, in Table 2, the ChebNetII baseline performs very well and is not that far off PolyAttn (Cheb). In fact, it outperforms PolyAttn (Mono) on some tasks. This is a bit disappointing since PolyAttn (Mono) is touted as being interpretable, but PolyAttn (Cheb) is the method that outperforms baselines.
>
> **A2:** (1) ''ChebNetII performs well and is not far off polyattn(cheb).''
>
> Below are the $R^2$ scores / the sum of squared error (denoted as SE) results from Table 2. A higher $R^2$ score indicates better performance, while a lower SE indicates better performance as well. Both the $R^2$ score and SE of PolyAttn (Cheb) significantly outperform those of ChebNetII. Specifically, comparing ChebNetII, PolyAttn (Cheb) reduced SE by up to **3500 times** in the Low-high pass setting.

---

> > ### Author Response · Authors · 2023-11-15
> > **Reply to Reviewer 13YF (2)**
> >
> > | Model (5k para) | Mixed low-pass    | Mixed high-pass   | Mixed band-pass   | Mixed rejection-pass | Low&high-pass     | Band&rejection-pass |
> > | --------------- | ----------------- | ----------------- | ----------------- | -------------------- | ----------------- | ------------------- |
> > | GPRGNN          | 0.9978/0.9784     | 0.9806/0.7846     | 0.9088/1.2977     | 0.9962/1.8374        | 0.8499/35.3719    | 0.9876/13.4890      |
> > | PolyAttn (Mono) | *0.9994/0.2550*   | *0.9935/0.2631*   | 0.9030/1.3798     | 0.9971/1.4025        | *0.9997/0.0696*   | *0.9992/0.8763*     |
> > | ChebNetII       | 0.9980/0.8991     | 0.9811/0.7615     | *0.9492/0.7229*   | *0.9982/0.8610*      | 0.8494/35.4702    | 0.9870/14.1149      |
> > | PolyAttn (Cheb) | **0.9997/0.1467** | **0.9960/0.0148** | **0.9945/0.0782** | **0.9996/0.1949**    | **0.9999/0.0118** | **0.9999/0.0416**   |
> >
> > (2) ''CheNetII outperforms PolyAttn (Mono) on some tasks (in table 2).''
> >
> > We conducted experiments on synthetic datasets with GRPGNN and PolyAttn (Mono), which uses the Monomial basis, as well as ChebNetII and PolyAttn (Cheb), which uses the Chebyshev basis. It is observed that, **under the same basis, PolyAttn shows a significant improvement** over polynomial graph neural networks (PolyAttn (Mono) outperforms GPRGNN in 5 out of 6 datasets; PolyAttn (Cheb) outperforms ChebNetII in all 6 datasets).
> >
> > Next, we explain why ChebNetII outperforms PolyAttn (Mono) on some tasks. As indicated by [7], the Chebyshev basis is theoretically superior to the Monomial basis in fitting filter functions. Consequently, due to the inherent limitations of the Monomial basis, PolyAttn (Mono) exhibits a slight disadvantage when fitting complex filter functions such as Mixed band-pass filters and Mixed rejection-pass filters. **Nevertheless, PolyAttn (Mono) still significantly outperforms ChebNetII in the remaining four tasks.**
> >
> > **W3:** I may have missed this, but are there any experiments showing how PolyAttn scales to more than one PolyAttn layer?
> >
> > **A3:** In Section 4.1, to ensure fairness in comparing PolyAttn with polynomial filters, we conducted experiments with only a single layer of PolyAttn.
> >
> > In Section 4.2, we conducted experiments on real datasets using the Polyformer, which includes **1-4 layers** of PolyAttn. More detailed information about the layers can be found in Appendix D and the supplementary materials.
> >
> > We hope these responses are helpful and will gladly answer any additional questions you may have.
> >
> >
> >
> > References:
> >
> > [1] Krzysztof Marcin Choromanski, et al. Rethinking attention with performers. In ICLR, 2021.
> >
> > [2] Vijay Prakash Dwivedi and Xavier Bresson. A generalization of transformer networks to graphs.
> >
> > [3] Devin Kreuzer, et al. Rethinking graph transformers with spectral attention. In NeurIPS, pp. 21618–21629, 2021.
> >
> > [4] Ladislav Ramp´asek, et al. Recipe for a general, powerful, scalable graph transformer. In NeurIPS, pp. 14501–14515, 2022.
> >
> > [5] Eli Chien, et al. Adaptive universal generalized pagerank graph neural network. In ICLR, 2021.
> >
> > [6] Mingguo He, et al. Bernnet: Learning arbitrary graph spectral filters via bernstein approximation. In NeurIPS, pp. 14239–14251, 2021.
> >
> > [7] Mingguo He, et al. Convolutional neural networks on graphs with Chebyshev approximation, revisited. In NeurIPS, pp. 7264–7276, 2022.
> >
> > [8] Xiyuan Wang, et al. How powerful are spectral graph neural networks. In ICML, 2022.
> >
> > [9] Yuhe Guo, et al. Graph Neural Networks with Learnable and Optimal Polynomial Bases. In ICML, 2023.

---

> > > ### Comment · Reviewer_13YF · 2023-11-20
> > >
> > > >In Section 4.2, we conducted experiments on real datasets using the Polyformer, which includes 1-4 layers of PolyAttn. More detailed information about the layers can be found in Appendix D and the supplementary materials.
> > >
> > > Are the final hyperparameters chosen from the hyperparameter search listed in the paper?
> > >
> > > Otherwise thank you for your response, I am keeping my score.

---

> ### Author Response · Authors · 2023-11-21
> **Reply to Reviewer 13YF (3)**
>
> We sincerely thank you for your valuable feedback on our paper. The final hyperparameters were selected from the hyperparameter search space presented in our paper. In the revised draft, these final hyperparameters for our model are listed in Appendix D.3.

---

### Official Review · Reviewer_BLsH · 2023-10-31

**Soundness:** 2 fair
**Presentation:** 3 good
**Contribution:** 2 fair
**Rating:** 5
**Confidence:** 5

**Summary:**

This paper proposes a new Graph Transformer called PolyFormer which contains two main steps. In the first step, PolyFormer constructs polynomial tokens for each node. In the second step, PolyFormer leverages a tailored polynomial attention mechanism to learn the final node representations from the constructed polynomial tokens. Empirical results seem to demonstrate the effectiveness of PolyFormer on the node classification task.

**Strengths:**

1.	This paper introduces different polynomial bases to construct the token sequence for each node.
2.	This paper develops a new attention mechanism to learn node representations from the token sequences.
3.	Extensive experiments have been conducted to demonstrate the effectiveness of the proposed method.

**Weaknesses:**

1. The novelty of this paper seems to be limited.
2. More challenging datasets need to be adopted in the experiment.
3. Some experimental settings are not reasonable.

**Questions:**

1. The proposed PolyFormer actually combines many existing techniques. The Polynomial Token could be regarded as an extension of Hop2Token[1] with different propagation strategies. Moreover, utilizing tanh() to compute the attention score and the node-shared attention bias also have been proposed and successfully implemented in previous works [2] and [3] respectively.
2. More challenging datasets need to be added into the experiments to validate the effectiveness of the proposed PolyFormer, including Actor, Squirrel, Chameleon and ogb-products. The first three are challenging heterophilic datasets and the last one is the representative large-scale graph dataset. Note that, Squirrel and Chameleon should consider the filtered versions proposed in [4].
3. I notice that authors conduct the complexity comparison in Section 4.3. The authors keep the total number of parameters approximately same for each model. However, I think the settings of this comparison are not reasonable since it is not the true parameter setting for each model to achieve the best performance on the dataset. Compared to NAGphormer, the proposed PolyFormer introduces order-wise MLP to initialize the query and key matrix, which inevitably increase the training cost. I just wonder the truly training cost of each model on its optimal parameter setting.


[1] Chen et al. NAGphormer: A Tokenized Graph Transformer for Node Classification in Large Graphs. ICLR 2023.

[2] Bo et al. Beyond Low-frequency Information in Graph Convolutional Networks. AAAI 2021.

[3] Chien et al. Adaptive universal generalized pagerank graph neural network. ICLR 2022.

[4] Platonov et al. A critical look at the evaluation of GNNs under heterophily: are we really making progress? ICLR 2023.

---

> ### Author Response · Authors · 2023-11-15
> **Reply to Reviewer BLsH (1)**
>
> We greatly appreciate your insightful feedback!
>
> **W1/Q1:** The novelty of this paper seems to be limited. The proposed PolyFormer actually combines many existing techniques. The Polynomial Token could be regarded as an extension of Hop2Token with different propagation strategies. Moreover, utilizing tanh() to compute the attention score and the node-shared attention bias also have been proposed and successfully implemented in previous works respectively.
>
> **A1:** Our work is **motivated** by polynomial filters, a method that conveys both theoretical guarantees and good empirical performance in graph neural networks, particularly performs well in tasks like node classification. Despite their strong theoretical and empirical performance, polynomial filters are usually node-unified [1, 2, 3, 4, 5, 6], limiting their expressive power. It is challenging to extend these to node-wise filters directly. Fortunately, with our specially designed model, we achieve an equivalence to more expressive node-wise filters in the spectral domain. On the other hand, most existing graph transformers, including NAGphormer, add spectral information via Laplacian eigendecomposition into the spatial domain, which is costly. However, as our model arises directly from the spectral domain, it eliminates the need for this high-cost addition of spectral information, while still achieving outstanding performance due to the powerful expressive power of spectral filters.
>
> More specifically, our model differs from NAGphormer in various aspects:
>
> 1. **Node Token Design Perspective.**
>    As highlighted in our Introduction, our polynomial tokens are conceptualized from a spectral perspective, designed to convey spectral information through various polynomial bases, such as the Monomial basis, Chebyshev basis (used in our paper), and even the Bernstein basis (in response to Reviewer tq9c). In contrast, NAGphormer's spatially-oriented Hop2Token generates node tokens by aggregating information from different hops in the spatial domain. Even with the Monomial basis, our PolyAttn enables the model to capture information directly in the spectral domain, avoiding the high-cost process of adding additional spectral information, a step necessary in NAGphormer and other popular Graph Transformers [7, 8, 9].
> 2. **Elaborately Designed PolyAttn vs. Vanilla Self-Attention.**
>    NAGphormer employs vanilla self-attention for encoding tokens derived from Hop2Token. In contrast, we introduce specially designed PolyAttn for our proposed polynomial tokens. **While some design elements (e.g., tanh(), node-shared attention bias) have been used in prior work, it is important to note that our primary focus is not on innovating new activation functions for attention scores or creating novel attention biases.** Instead, our focus is on developing an attention mechanism specifically tailored for polynomial tokens, establishing a direct connection with polynomial filters in the spectral domain.
>
> With the polynomial tokens in the spectral domain and the tailored PolyAttn, our model effectively addresses two significant challenges in previous works:
>
> 1. **Graph Transformer models** that add spectral information (e.g., NAGphormer and  [7, 8, 9]) enhance expressive power but require extensive time and space. In contrast, our model, which originates from the spectral domain, already possesses powerful expressive capabilities. Without the need for additional spectral information, which incurs significant costs, our model demonstrates great scalability and performance, capable of handling graphs with up to **100 million nodes**, as illustrated in Experiment 4.2.
> 2. **Polynomial filters** (e.g., [1, 2, 3, 4, 5, 6]) are typically node-unified, learning the same filter for all nodes and thereby limiting their expressive power. Our model operates as a **node-wise** filter, adaptively learning distinct filters for different nodes, as theoretically substantiated in Theorem 3.1 and Proposition 3.1, and empirically demonstrated in Experiment 4.1.
>
> In summary, inspired by the concept of spectral filters, our approach uniquely crafts node tokens and attention mechanisms from a spectral viewpoint, contrasting with NAGphormer's spatial focus. Leveraging polynomial tokens and PolyAttn, our model not only theoretically aligns with but also empirically aligns the expressive node-wise filters. These properties help to address challenges in both Graph Transformers and polynomial filters. Based on this elaborate design, our model offers a desirable balance between scalability and expressiveness.

---

> > ### Author Response · Authors · 2023-11-15
> > **Reply to Reviewer BLsH (2)**
> >
> > **W2/Q2:** More challenging datasets need to be adopted in the experiment. More challenging datasets need to be added into the experiments to validate the effectiveness of the proposed PolyFormer, including Actor, Squirrel, Chameleon and ogb-products. The first three are challenging heterophilic datasets and the last one is the representative large-scale graph dataset. Note that, Squirrel and Chameleon should consider the filtered versions.
> >
> > **A2:** For the Actor dataset, we employed a 60%/20%/20% split for training, validation, and testing, respectively. For the other three datasets, we utilized the provided splits. Below are the corresponding results on 10 runs.
> >
> > |                   | Actor      | Chameleon (Filtered) | Squirrel (Filtered) | ogbn-products |
> > | :---------------- | ---------- | -------------------- | ------------------- | :------------ |
> > | GPRGNN            | 40.41±0.72 | 42.28±2.87           | 41.09±1.18          | 76.00±0.16    |
> > | ChebNetII         | 42.05±0.97 | 42.67±1.43           | 41.22±0.37          | 75.05±0.14    |
> > | Specformer        | 42.24±0.93 | 42.82±2.54           | 40.60±0.53          | -             |
> > | NAGphormer        | 40.88±1.37 | 40.36±1.77           | 39.79±0.84          | -             |
> > | PolyFormer (Mono) | 42.29±0.74 | 46.86±1.61           | 42.56±0.96          | 76.97±0.16    |
> > | PolyFormer (Cheb) | 43.29±0.80 | 45.35±2.97           | 41.83±1.18          | 76.93±0.40    |
> >
> > According to the table above, PolyFormer outperforms the baseline models on these challenging heterophilic datasets and large-scale datasets.

---

> > > ### Author Response · Authors · 2023-11-15
> > > **Reply to Reviewer BLsH (3)**
> > >
> > > **W3/Q3:** Some experimental settings are not reasonable. I notice that authors conduct the complexity comparison in Section 4.3. The authors keep the total number of parameters approximately same for each model. However, I think the settings of this comparison are not reasonable since it is not the true parameter setting for each model to achieve the best performance on the dataset. Compared to NAGphormer, the proposed PolyFormer introduces order-wise MLP to initialize the query and key matrix, which inevitably increase the training cost. I just wonder the truly training cost of each model on its optimal parameter setting.
> > >
> > > **A3:** The comparison of model complexity, given a similar parameter count, **was conducted in line with the settings used in prior works** [1, 2, 10]. However, evaluating the cost-effectiveness at optimal performance settings is also a critical aspect. To address this, we extended complexity comparison experiments under the optimal conditions on the dataset Roman-empire. The results are summarized as follows:
> > >
> > > | Model       | Acc.       | Prep. Time(s) | Train. (ms/epoch) | Infer. (ms/epoch) | Max GPU Mem. (MB) |
> > > | ----------- | ---------- | ------------- | ----------------- | ----------------- | ----------------- |
> > > | Transformer | 65.29±0.47 | 0             | 1721              | 739               | 11457             |
> > > | NAGphormer  | 74.45±0.48 | 42.12         | 426               | 214               | 3063              |
> > > | Specformer  | 69.94±0.34 | 1120.73       | 3931              | 7233              | 42813             |
> > > | PolyFormer  | 78.89±0.39 | 0.24          | 1079              | 733               | 3319              |
> > >
> > > In optimal settings, PolyFormer exhibits a marginally slower training and inference time compared to NAGphormer, which may be attributable to design choices such as order-wise MLPs. However, given that PolyFormer **significantly outperforms** NAGphormer in preprocessing speed and demonstrates considerable performance improvements, this trade-off is deemed acceptable. Regarding GPU memory overhead, PolyFormer and NAGphormer are virtually identical. It is important to note that our evaluations were conducted under full batch settings. Theoretically, a reduced GPU overhead is attainable with minibatch-based processing. Overall, PolyFormer stands as a highly competitive option, considering accuracy, preprocessing speed, training and inference times, and GPU memory requirements.
> > >
> > >
> > >
> > > References:
> > >
> > > [1] Eli Chien, et al. Adaptive universal generalized pagerank graph neural network. In ICLR, 2021.
> > >
> > > [2] Mingguo He, et al. Bernnet: Learning arbitrary graph spectral filters via bernstein approximation. In NeurIPS, pp. 14239–14251, 2021.
> > >
> > > [3] Mingguo He, et al. Convolutional neural networks on graphs with Chebyshev approximation, revisited. In NeurIPS, pp. 7264–7276, 2022.
> > >
> > > [4] Xiyuan Wang, et al. How powerful are spectral graph neural networks. In ICML, 2022.
> > >
> > > [5] Yuhe Guo, et al. Graph Neural Networks with Learnable and Optimal Polynomial Bases. In ICML, 2023.
> > >
> > > [6] Bo, Deyu, et al. Beyond low-frequency information in graph convolutional networks. In AAAI 2021.
> > >
> > > [7] Devin Kreuzer, et al. Rethinking graph transformers with spectral attention. In NeurIPS, pp. 21618–21629, 2021.
> > >
> > > [8] Ladislav Ramp´asek, et al. Recipe for a general, powerful, scalable graph transformer. In NeurIPS, pp. 14501–14515, 2022.
> > >
> > > [9] Deyu Bo, et al. Specformer: Spectral graph neural networks meet transformers. In ICLR, 2023.
> > >
> > > [10] Qitian Wu, et al. Simplifying and Empowering Transformers for Large-Graph Representations. In NeurIPS, 2023.

---

> > > > ### Author Response · Authors · 2023-11-23
> > > > **A Kind Reminder**
> > > >
> > > > Dear reviewer BLsH,
> > > >
> > > > Thank you very much for your review. We hope that you will find our responses satisfactory and that they help clarify your concerns. We appreciate the opportunity to engage with you. We would like to kindly remind you that the discussion period is coming to an end. Could you please inform us if our responses have resolved your concerns or if there are any other questions you need us to address? Thank you.

---

> > > > > ### Comment · Reviewer_BLsH · 2023-11-23
> > > > >
> > > > > Thanks for the detailed responses.
> > > > > My opinion of the paper remains unchanged and I will keep my score.

---

### Official Review · Reviewer_tq9c · 2023-11-01

**Soundness:** 3 good
**Presentation:** 4 excellent
**Contribution:** 3 good
**Rating:** 6
**Confidence:** 3

**Summary:**

The paper introduces Polyformer, a novel graph neural network architecture designed to balance scalability and expressiveness for node-level prediction tasks on graphs. It innovatively introduces node tokens based on polynomial bases to efficiently capture node neighborhood information, which not only allows for minibatch training but also enhances scalability. Furthermore, the paper proposes a Polynomial Attention (PolyAttn) mechanism specifically designed for these polynomial node tokens. PolyAttn serves as a node-wise spectral filter, offering more expressive power than the node-unified filters used in previous works. The Polyformer architecture is built using these polynomial node tokens and the PolyAttn mechanism. Experimental results demonstrate that Polyformer outperforms previous state-of-the-art models in node classification tasks, effectively handling graphs with up to 100 million nodes.

**Strengths:**

- Novel node token formulation using polynomial bases, enabling scalability.
- PolyAttn provides node-wise filtering and enhanced expressiveness.
- Strong experimental results demonstrating scalability and performance.

**Weaknesses:**

- The polynomial bases are somewhat limited to Monomial and Chebyshev. More advanced bases could be explored.
- Comparisons to very recent graph neural network methods are missing.

**Questions:**

- How does PolyAttn relate to self-attention in standard Transformers? Is it a specialized version?
- For large graphs, is recomputing the polynomial tokens for each minibatch really efficient?

---

> ### Author Response · Authors · 2023-11-15
> **Reply to Reviewer tq9c**
>
> Thank you very much for your valuable feedback!
>
> **W1:** The polynomial bases are somewhat limited to Monomial and Chebyshev. More advanced bases could be explored.
>
> **A1:** Your suggestion is constructive. In fact, our model framework can be extended to other polynomial bases. Accordingly, we have included results for PolyFormer with the Bernstein basis [1] and the optimal basis [2]. The specific results are as follows:
>
> |              | GPRGNN     | ChebNetII  | NAGphormer | Specformer | PolyFormer (Bern) | PolyFormer (Opt) | PolyFormer (Mono) | PolyFormer (Cheb) |
> | ------------ | ---------- | ---------- | ---------- | ---------- | ----------------- | ---------------- | ----------------- | ----------------- |
> | Tolokers     | 77.25±0.61 | 76.83±0.53 | 81.57±0.44 | 80.42±0.55 | 84.32±0.59        | 83.15±0.49       | 84.00±0.45        | 83.88±0.33        |
> | Roman-empire | 74.08±0.54 | 72.70±0.30 | 74.45±0.48 | 69.94±0.34 | 77.64±0.33        | 77.15±0.33       | 78.89±0.39        | 80.27±0.39        |
>
> Compared to the baseline models, PolyFormer models based on different basis all perform well.
>
> **W2:** Comparisons to very recent graph neural network methods are missing.
>
> **A2:** We have supplemented with the latest Graph Transformer model GOAT [3] to enrich our baseline. The specific results are as follows:
>
> |                   | CS         | Physics    | Tolokers   | Roman-empire |
> | ----------------- | ---------- | ---------- | ---------- | ------------ |
> | GPRGNN            | 95.26±0.15 | 97.84±0.59 | 77.25±0.61 | 74.08±0.54   |
> | ChebNetII         | 96.33±0.12 | 97.25±0.78 | 79.23±0.43 | 74.64±0.39   |
> | Specformer        | 96.07±0.10 | 97.70±0.60 | 80.42±0.55 | 69.94±0.34   |
> | NAGphormer        | 94.41±0.35 | 97.18±0.36 | 81.57±0.44 | 74.45±0.48   |
> | GOAT              | 95.12±0.21 | 97.09±0.24 | 83.26±0.52 | 72.30±0.48   |
> | PolyFormer (Mono) | 96.30±0.14 | 98.16±0.28 | 83.66±0.63 | 78.47±0.38   |
> | PolyFormer (Cheb) | 96.47±0.14 | 98.01±0.37 | 83.41±0.47 | 79.62±0.22   |
>
> **Q1:** How does PolyAttn relate to self-attention in standard Transformers? Is it a specialized version?
>
> **A3:** PolyAttn can be considered as a specially designed version of self-attention tailored for polynomial tokens. Its primary objective is to establish a connection between attention mechanisms and polynomial filters, as illustrated in Theorem 3.1.
>
> Specifically, the design of PolyAttn incorporates elements such as $\tanh()$ activation function and order-wise MLPs. Detailed information about its design can be found in Algorithm 1 of our paper. Using vanilla self-attention in conjunction with polynomial tokens would be insufficient to achieve equivalence with polynomial filters or may restrict the filtering capabilities. For instance, vanilla attention with polynomial tokens possibly only act as low-pass filters. These were discussed in Proposition 3.2 of our paper. This limitation would result in a reduction of expressive power.
>
> Therefore, to enrich model expressiveness and ensure theoretical guarantee, we designed a specialized version of self-attention, PolyAttn, for polynomial tokens.
>
> **Q2:** For large graphs, is recomputing the polynomial tokens for each minibatch really efficient?
>
> **A4:** During the preprocessing stage before training and inference, our model precomputes and stores polynomial tokens for all nodes.
>
> Once training and inference begin, these precomputed polynomial tokens are directly loaded for each minibatch, **eliminating the need for their recomputation．**
>
> From the preprocessing stage to the phases of training and inference, **polynomial tokens for all nodes are computed only once.** Consequently, this approach is efficient for large graphs in practical applications.
>
> We hope our answers are helpful. We are looking forward to your positive feedback.
>
>
>
> References:
>
> [1] Mingguo He, Zhewei Wei, Zengfeng Huang, and Hongteng Xu. Bernnet: Learning arbitrary graph spectral filters via bernstein approximation. In NeurIPS, pp. 14239–14251, 2021.
>
> [2] Yuhe Guo, and Zhewei Wei. Graph Neural Networks with Learnable and Optimal Polynomial Bases. In ICML, 2023.
>
> [3] Kong, Kezhi and Chen, Jiuhai and Kirchenbauer, John and Ni, Renkun and Bruss, C Bayan and Goldstein, Tom. GOAT: A Global Transformer on Large-scale Graphs. In ICML, 2023.

---

> > ### Author Response · Authors · 2023-11-23
> > **A Kind Reminder**
> >
> > Dear reviewer tq9c,
> >
> > Thank you very much for your review. We hope that you will find our responses satisfactory and that they help clarify your concerns. We appreciate the opportunity to engage with you. We would like to kindly remind you that the discussion period is coming to an end. Could you please inform us if our responses have resolved your concerns or if there are any other questions you need us to address? Thank you.

---

### Official Review · Reviewer_Wuvt · 2023-11-08

**Soundness:** 2 fair
**Presentation:** 3 good
**Contribution:** 3 good
**Rating:** 5
**Confidence:** 5

**Summary:**

Overall, the research problem is interesting and the research topic of graph transformer is novel. Furthermore, the proposed method PolyFormer method is clearly presented with theoretical analysis, and the experimental results are promising. On the other side, some concerns are raised. Please refer to the following sections for details.

**Strengths:**

- The research problem is interesting. Trying to reduce the computational complexity and maintaining the effectiveness is pragmatic.

- The proposed method PolyFormer is easy to understand and complete with theoretical analysis.

- The experimental setting is extensive and results are competitive.

**Weaknesses:**

- The introduction of NAGphormer seems incorrect somehow. On the one hand, the authors mention that NAGphormer was designed based on spatial information and neglected spectral information. On the other hand, the authors mention that NAGphormer attempted to use spectral information but eigendecomposition is costly. It seems contradictory. Moreover, if PolyFormer uses the proposed Monomial Basis especially, it seems that PolyFormer and NAGphormer are under the same general framework. The reviewer would appreciate if the authors could address this concern during the rebuttal.

- The novelty is incremental compared with NAGphormer, PolyFormer adds MLP on each hop aggregation of NAGphormer, to some extent.

- Theorem 1 seems not informative somehow.

**Questions:**

Extending the first bullet point in the weaknesses section, could the authors explain why the proposed "polynomial token" with PolyAttn is a spectral method, not a spatial method, based on the polynomial type listed in Table 1?

How to understand $h(\lambda)$ in Figure 1 with $\alpha$ in Eq. 8? Moreover, during the learning process, how to obtain and interpret the low-pass and high-pass?

---

> ### Author Response · Authors · 2023-11-15
> **Reply to Reviewer Wuvt (1)**
>
> We are very grateful for your valuable and constructive feedback!
>
> **W1:** The introduction of NAGphormer seems incorrect somehow. Moreover, if PolyFormer uses the proposed Monomial Basis especially, it seems that PolyFormer and NAGphormer are under the same general framework.
>
> **A1:**  **(1) The clarification about NAGphormer.**
>
> NAGphormer mainly comprises two parts:
>
> *a. Hop2Token with vanilla multi-head self-attention.* NAGphormer generates node tokens via aggregating neighbors' information across different hops (Hop2Token) and encodes these tokens via vanilla multi-head self-attention. **This process is designed on the spatial domain** and neglects the spectral information, as discussed in the first paragraph of the Introduction section.
>
> *b. Structural encoding.* To include spectral information, NAGphormer uses eigenvectors derived from Laplacian eigendecomposition in the model's implementation. NAGphormer calls this technique the structural encoding, which is costly. We mentioned this in the second paragraph of our Introduction section.
>
> **(2) Different framework between PolyFormer and NAGphormer.**
>
> Firstly, we would like to emphasize that our work is **motivated** by node-wise filters. Polynomial filters have been proven to be a successful spectral method in graph neural networks and perform well in tasks such as node classification. However, most of them are node-unified filters, limiting the enhancement of expressive power. It is challenging to expand polynomial filters to node-wise filters directly. Fortunately, with the aid of attention mechanisms, we establish an equivalence between our model and node-wise filters.
>
> Based on the motivation above, we clarify that our proposed model operates within a **different framework** compared to NAGphormer. As we have claimed, one of our aims is to establish a connection between our model and expressive node-wise filters in the spectral domain. We design node tokens with the assistance of polynomial bases, which contain spectral information, especially when combined with our proposed PolyAttn. However, Hop2Token derives node tokens by aggregating different hop information in the spatial domain. The Monomial basis is a special case that only superficially makes our model appear similar to NAGphormer. Nonetheless, it turns out that PolyFormer is distinct from NAGphormer in terms of other bases (such as the Chebyshev basis in our paper, or the Bernstein basis and the optimal basis in response to Reviewer tq9c). Even when considering the Monomial basis, PolyFormer differentiates from NAGphormer as its PolyAttn with Monomial tokens enables the model to act as node-wise spectral filters, while the vanilla attention used in NAGphormer with tokens from Hop2Token cannot. Due to the superior spectral property in PolyFormer, our model generates a more difference in framework compared to NAGphormer: NAGphormer requires the addition of spectral information as a supplement, which incurs high costs, while our work effectively and directly captures spectral information, **eliminating this need.**
>
> In summary, inspired by the spectral node-wise filter, our model integrates spectral information through polynomial tokens and PolyAttn, operating under a distinct framework compared to NAGphormer, which is designed from the spatial perspective.
>
> **W2:** The novelty is incremental compared with NAGphormer, PolyFormer adds MLP on each hop aggregation of NAGphormer, to some extent.
>
> **A2:** We distinguish our work from NAGphormer based on motivation, model design, and theory and empirical performance.
>
> **(1) Motivation.** Inspired by the strong performance and theoretical guarantees of spectral filters, we aim to build a bridge between Graph Transformers and expressive spectral filters, a feat not achieved by NAGphormer. Our design of polynomial tokens and PolyAttn even establishes an equivalence to more expressive node-wise filters.
>
> **(2) Detailed Design Differences between PolyFormer and NAGphormer.**
>
> *a. Spectral-based Node Tokens.* NAGphormer's Hop2Token aggregates information from different hops in the spatial domain. In implementation, NAGphormer necessitates additional structural encoding based on Laplacian eigendecomposition to supplement spectral information, leading to high costs. Conversely, our polynomial tokens, designed in the spectral domain, inherently contain spectral information, obviating the need for additional positional encoding.

---

> > ### Author Response · Authors · 2023-11-15
> > **Reply to Reviewer Wuvt (2)**
> >
> > *b. Specialized PolyAttn.* Unlike NAGphormer, which uses vanilla self-attention for tokens from Hop2Token, we have tailored PolyAttn for polynomial tokens. This customization in our approach goes beyond incorporating order-wise MLPs. We intricately design PolyAttn, including order-wise MLPs, tanh() activation to enable both positive and negative attention scores, and aligning the Value matrix to the polynomial basis. This establishes an equivalence between our model and polynomial filters in the spectral domain, endowing the Polyformer with advanced expressive capabilities compared to NAGphormer, which is spatially oriented.
> >
> > **(3) Theory and Empirical Performance.** In Section 3.3, we theoretically demonstrate that our model acts as an expressive node-wise filter. Furthermore, we analyze how multi-head PolyAttn functions as multi-channel filters, which are more expressive. These properties are not possessed by NAGphormer. In addition, we analyze how the vanilla attention in NAGphormer could limit the expressive power. In Section 4.1, we empirically prove that our model can learn node-wise filters on both synthetic and real-world datasets, outperforming NAGphormer. Specifically, the proposed Graph Transformer PolyFormer outperforms NAGphormer on homophilic, heterophilic, and large-scale graphs.
> >
> > In conclusion, motivated by spectral filters, we directly design node tokens and attention from a spectral perspective, whereas NAGphormer is mainly spatially oriented. With the help of polynomial tokens and PolyAttn, we both theoretically and empirically demonstrate that our model achieves equivalence to expressive node-wise filters. Without the need for additional spectral supplementation, which is implemented in NAGphormer, our model outperforms baselines on various datasets.
> >
> > **W3:** Theorem 1 (i.e. Theorem 3.1) seems not informative somehow.
> >
> > **A3:** As mentioned in A2, we established a connection between our model and polynomial filters based on the proposed polynomial tokens and PolyAttn. It's important to note that most existing polynomial GNNs [1, 2, 3, 4, 5] learn a **node-unified** filter, i.e., all nodes share the same filter, limiting the model's expressiveness. For instance, when nodes follow different filter patterns (e.g., high-pass for some nodes and low-pass for others), node-unified filters struggle to match such patterns. In contrast, our model acts as a **node-wise** filter. Specifically, Theorem 3.1 **theoretically constructs the bridge between PolyAttn** ($\sum\_{k=0}^{K} \mathbf{H}'^{(i)}\_{k,:}$ in Theorem 3.1) **and the node-wise filter** ($\sum\_{k=0}^K \alpha^{(i)}\_k (g\_k{(\mathbf{P})} \mathbf{X})\_{i,:}$ in Theorem 3.1).
> >
> > **Q1:** Extending the first bullet point in the weaknesses section, could the authors explain why the proposed "polynomial token" with PolyAttn is a spectral method, not a spatial method, based on the polynomial type listed in Table 1?
> >
> > **A4:** We further elaborate based on Theorem 3.1. Combining equations (8) and (2) from the paper, we obtain:
> >
> > $\mathbf{Z}\_{i,:} = \sum\_{k=0}^K \mathbf{H}'^{(i)}\_{k,:} =\sum_{k=0}^K \alpha_k^{(i)}(g\_k(\mathbf{P})\mathbf{X})\_{i,:} = \left(\mathbf{U}h^{(i)}(\mathbf{\Lambda})\mathbf{U}^\top \mathbf{X}\right)\_{i,:}$.
> >
> > Here, $\mathbf{Z}\_{i,:}$ is the representation of the node, $\mathbf{H}'^{(i)}\_{k,:}$ is the representation obtained by polynomial token via PolyAttn, $\mathbf{U}$ and $\mathbf{\Lambda}$ are the eigenvectors and eigenvalues corresponding to the Laplacian matrix respectively, $\mathbf{X}$ is the original representation of the node. This illustrates that the node representation learned with polynomial tokens via PolyAttn is derived in the spectral domain, i.e. $\mathbf{Z}\_{i,:} = \left(\mathbf{U}h^{(i)}(\mathbf{\Lambda})\mathbf{U}^\top \mathbf{X}\right)\_{i,:}$.
> >
> > For different types of polynomial tokens used, as shown in Table 1, the expression of $ h(\mathbf{\Lambda}) $ varies. However, the learning of node representation can consistently be expressed in the form of the formula above [1, 3], indicating learning in the spectral domain.

---

> > > ### Author Response · Authors · 2023-11-15
> > > **Reply to Reviewer Wuvt (3)**
> > >
> > > **Q2:** How to understand $h(\lambda)$ in Figure 1 with $\alpha$ in Eq. 8? Moreover, during the learning process, how to obtain and interpret the low-pass and high-pass?
> > >
> > > **A5:** We answer the question from two aspects.
> > >
> > > **(1) The relationship between $ h(\lambda) $ in Figure 1 and $\alpha$ in Eq. 8.**
> > >
> > > In Fig. 1, $ h(\lambda) $ denotes a mapping function applied to the eigenvalues of the normalized Laplacian matrix, where $\lambda \in [0,2]$. We define $ h^{(i)}(\lambda) $ as the function corresponding to node $ i $. In Eq. 8, $\alpha_k^{(i)}$, for $ k=\{0,\cdots, K\} $, represents the set of polynomial coefficients for node $ i $. The relationship between $ h^{(i)}(\lambda) $ and $\alpha_k^{(i)}$ is determined by the choice of polynomial tokens:
> > >
> > > * Monomial basis: $ h^{(i)}(\lambda) = \sum_{k=0}^{K}\alpha_k^{(i)} \cdot (1-\lambda)^k $
> > > * Chebyshev basis: $ h^{(i)}(\lambda) = \sum_{k=0}^{K} \alpha_k^{(i)} \cdot T_k(\lambda) $, where $ T_k(\lambda) $ denotes the Chebyshev polynomial of $ T_0(\lambda) = 1, T_1(\lambda) = \lambda - 1 $, and $ T_k(\lambda) = 2(\lambda - 1) \cdot T_{k-1}(\lambda) - T_{k-2}(\lambda) $ for $ k \geq 2 $
> > >
> > > **(2) Interpreting and obtaining the low-pass and high-pass filters.**
> > >
> > > In simpler terms, a low-pass filter $ h(\lambda) $ preserves smaller eigenvalues $\lambda$ of the Laplacian and attenuates larger ones, while a high-pass filter does the opposite. For instance, $ h(\lambda) = 1 - \frac{\lambda}{2}, \lambda \in [0,2] $ is an example of a linear low-pass filter, while $ h(\lambda) = \frac{\lambda}{2} $ exemplifies a linear high-pass filter. Low-pass filtering tends to smooth the node signals in the graph, whereas high-pass filtering emphasizes the differences in signals [6].
> > >
> > > During the learning process, filters are obtained by learning the trainable parameters in PolyAttn (e.g. $\mathbf{W}_Q$ and $\mathbf{W}_K$). This leads to the generation of appropriate attention scores and, subsequently the acquisition of polynomial filter coefficients. The detailed relationship between the attention scores in PolyAttn and the polynomial filter coefficients is illustrated in the proof of Theorem 3.1. With the acquisition of polynomial filter coefficients $\alpha_k^{(i)}$ for node $i$, we have the corresponding filter $h^{(i)}(\lambda)$ .
> > >
> > > Furthermore, we verify that our model successfully learns effective filters on both synthetic and real-world datasets, as discussed in sections 4.1.1 and 4.1.2. Notably, our model is capable of learning distinct filters for different nodes, effectively acting as a node-wise filter. For example, Figure 2(b) illustrates that our model learns **low-pass** filters for certain nodes and **high-pass** filters for others in a synthetic dataset, yielding optimal results as indicated in Table 2. Similarly, on real-world datasets, our model adapts by learning diverse filters for each node, demonstrating more expressive capabilities compared to node-unified filters, as presented in Table 3.
> > >
> > > In summary, our model is capable of both theoretically and empirically learning diverse filters for nodes (e.g., low-pass filter, high-pass filter).
> > >
> > > We hope these responses are helpful and will gladly answer any additional questions you may have.
> > >
> > >
> > >
> > > References:
> > >
> > > [1] Eli Chien, Jianhao Peng, Pan Li, and Olgica Milenkovic. Adaptive universal generalized pagerank graph neural network. In ICLR, 2021.
> > >
> > > [2] Mingguo He, Zhewei Wei, Zengfeng Huang, and Hongteng Xu. Bernnet: Learning arbitrary graph spectral filters via bernstein approximation. In NeurIPS, pp. 14239–14251, 2021.
> > >
> > > [3] Mingguo He, Zhewei Wei, and Ji-Rong Wen. Convolutional neural networks on graphs with Chebyshev approximation, revisited. In NeurIPS, pp. 7264–7276, 2022.
> > >
> > > [4] Xiyuan Wang, and Muhan Zhang. How powerful are spectral graph neural networks. In ICML, 2022.
> > >
> > > [5] Yuhe Guo, and Zhewei Wei. "Graph Neural Networks with Learnable and Optimal Polynomial Bases." In ICML, 2023.
> > >
> > > [6] Elvin Isufi, Fernando Gama, David I. Shuman, and Santiago Segarra. Graph filters for signal processing and machine learning on graphs.

---

> > > > ### Author Response · Authors · 2023-11-23
> > > > **A Kind Reminder**
> > > >
> > > > Dear reviewer Wuvt,
> > > >
> > > > Thank you very much for your review. We hope that you will find our responses satisfactory and that they help clarify your concerns. We appreciate the opportunity to engage with you. We would like to kindly remind you that the discussion period is coming to an end. Could you please inform us if our responses have resolved your concerns or if there are any other questions you need us to address? Thank you.

---

> > > > > ### Comment · Reviewer_Wuvt · 2023-11-23
> > > > >
> > > > > Thanks for the response, I have no further questions. The paper will be discussed among reviewers and AC.

---

### Author Response · Authors · 2023-11-20
**Summary of the Updates during Rebuttal**

We sincerely thank all the reviewers for their valuable and constructive feedback. Here, we would like to summarize our updates during the rebuttal period as follows.

**In the Reply to reviewer Wuvt:**

- We present a distinct framework comparison between PolyFormer and NAGphormer, covering aspects such as motivation, detailed design differences, and theory and empirical performance comparisons.
- We place emphasis on Theorem 1 (i.e., Theorem 3.1) as a crucial link between PolyAttn and the node-wise filter, which furnishes our model with essential theoretical guarantees.
- We further elucidate the relationship between node-wise filter coefficients $\alpha$ and spectral filter $h(\lambda)$, both with Monomial basis and Chebyshev basis, and explain how the model can learn different filters for each node (e.g., low-pass filter and high-pass filter).

**In the Reply to reviewer tq9c:**

- We explore more advanced bases (Bernstein basis and the optimal basis) and include additional recent baselines. Our model continues to achieve outstanding performance on real-world datasets.
- We highlight the differences between our specially designed PolyAttn and vanilla self-attention, which theoretically guarantee our model's increased expressiveness.
- We emphasize that there is no need to recompute the polynomial tokens for each minibatch, ensuring the high efficiency of our model.

**In the Reply to reviewer BLsH:**

- We underscore the novelty of our approach, stemming from the motivation drawn from spectral filters and the unique design of node token and attention mechanisms, which significantly differ from NAGphormer. With the tailored design in the spectral domain, our model effectively addresses two significant challenges in both Graph Transformer works and Polynomial filters.
- We conduct experiments on more challenging datasets, including Actor, Chameleon (Filtered), Squirrel (Filtered), and ogbn-products, demonstrating the outstanding performance of our models on these heterophilic datasets and large-scale datasets.
- Additionally, we provide a complexity comparison in the optimal performance setting. PolyFormer stands as a highly competitive option, considering accuracy, preprocessing speed, training and inference times, and GPU memory requirements.

**In the Reply to reviewer 13YF:**

- We explain the advantages of choosing intra-node attention in our model, both in terms of scalability and model performance.
- We clarify that our model significantly outperforms baselines in most experiments and provide explanations for these results.
- We clarify the multiple layers of PolyAttn, showcasing the scalability of our model.

We will carefully review the writing of our paper. If you have any further questions, we welcome further discussion.

Thanks for your time and efforts in reviewing.

---

### Meta-Review · Area_Chair_d2ep · 2023-12-06

**Metareview:**

This work introduces Polyformer, a transformer-based graph neural network architecture for node-level prediction tasks on graphs. The key insight is replacing the spectral positional embedding with a feature propagation one ($g_k(P)$) and intra-node attention. The graph structure enters only through the feature propagation in the "location encoding", and in a way that heavily depends on the node features. Moreover, there is no exchange of information between the nodes after the "location encoding" phase. The authors' rebuttal argument that this is good enough for their tasks may also mean that their tasks are too easy.

- There are some good ideas in the paper (e.g., use of tanh activation, scalability through intra-node attention) but one would hope graph structure information in the final embedding would be richer (it works on some tasks but it may be because they are easy).

- It is unclear why $g_k(P)$ would not suffer from oversquashing and oversmoothing.

**Justification For Why Not Higher Score:**

This approach is scalable but sacrifices encoding graph structure too much.

**Justification For Why Not Lower Score:**

N/A

---

### Decision · Program_Chairs · 2024-01-16

Reject